# FAB-PPI: Frequentist, Assisted by Bayes, Prediction-Powered Inference

**Stefano Cortinovis** [1]  **François Caron** [1]

## Abstract

Prediction-powered inference (PPI) enables valid statistical inference by combining experimental data with machine learning predictions. When a sufficient number of high-quality predictions is available, PPI results in more accurate estimates and tighter confidence intervals than traditional methods. In this paper, we propose to inform the PPI framework with prior knowledge on the quality of the predictions. The resulting method, which we call frequentist, assisted by Bayes, PPI (FAB-PPI), improves over PPI when the observed prediction quality is likely under the prior, while maintaining its frequentist guarantees. Furthermore, when using heavy-tailed priors, FAB-PPI adaptively reverts to standard PPI in low prior probability regions. We demonstrate the benefits of FAB-PPI in real and synthetic examples.

## 1. Introduction

Statistical inference crucially relies on the availability of high-quality labelled data to draw actionable conclusions. As the scale of machine learning models keeps growing, their increasingly accurate predictions become a tempting alternative to labelled data in fields where the latter are traditionally scarce, such as proteomics (Jumper et al., 2021). However, blindly using potentially biased predictions as a surrogate for labelled data voids the statistical validity of the conclusions drawn. To address this, prediction-powered inference (Angelopoulos et al., 2023a) provides a general framework for statistical inference in the presence of a large number of black-box predictions by combining them with a smaller number of labelled observations, which are used to *correct* for the discrepancy between the predictions and the true labels. The estimators and confidence intervals (CIs) resulting from PPI are statistically valid regardless of the machine learning model used. Moreover, when the predictions

are good, PPI results in more accurate estimates and shorter CIs than traditional methods that rely solely on labelled data.

More formally, for an input/output pair $(X, Y) \sim \mathbb{P} = \mathbb{P}_X \times \mathbb{P}_{Y|X}$ and a convex loss function $\mathcal{L}_\theta(x, y)$, where $\theta \in \mathbb{R}^d$, we wish to estimate

$$\theta^\star = \underset{\theta \in \mathbb{R}^d}{\arg\min} \ \mathbb{E}[\mathcal{L}_\theta(X, Y)]. \quad (1)$$

For instance, if $\mathcal{L}_\theta(x, y) = (\theta - y)^2/2$ is the squared loss, then $\theta^\star = \mathbb{E}[Y]$. We assume that we have $n$ labelled observations $\{(X_i, Y_i)\}_{i=1}^n$ iid from $\mathbb{P}$ and $N$ unlabelled observations $\{\widetilde{X}_i\}_{i=1}^N$ iid from $\mathbb{P}_X$, which are also independent of the labelled data. The number of unlabelled observations is typically much larger than the number of labelled ones, $N \gg n$. Additionally, we are provided with a machine learning prediction rule $f$, that can be used to predict an output $f(x)$ at any input $x$. PPI aims to obtain an estimator $\widehat{\theta}$ and a $(1 - \alpha)$ confidence interval $\mathcal{C}_\alpha^{\mathrm{pp}}$ for $\theta^\star$, which take advantage of $f$. Under mild assumptions, $\theta^\star$ can be expressed as the solution to

$$g_{\theta^\star} := \mathbb{E}[\mathcal{L}'_{\theta^\star}(X, Y)] = 0, \quad (2)$$

where $\mathcal{L}'_\theta$ is a subgradient of $\mathcal{L}_\theta$ with respect to $\theta$. It is easy to see that the quantity above can be decomposed as $g_\theta = m_\theta + \Delta_\theta$, where

$$m_\theta := \mathbb{E}[\mathcal{L}'_\theta(X, f(X))], \quad (3)$$
$$\Delta_\theta := \mathbb{E}[\mathcal{L}'_\theta(X, Y) - \mathcal{L}'_\theta(X, f(X))]. \quad (4)$$

In this setting, $m_\theta$ represents a measure of fit of the predictor, whereas $\Delta_\theta$, called the *rectifier*, accounts for the discrepancy between the predicted outputs $f(X)$ and the true outputs $Y$, effectively quantifying prediction quality. For example, under the squared loss, $\Delta_\theta = \mathbb{E}[f(X) - Y]$ and a *good* predictor $f$ is one such that $\Delta_\theta$ is close to zero, i.e. $f(x) \simeq \mathbb{E}[Y|X = x]$. Note that, while in this case $\Delta_\theta$ does not depend on $\theta$, this is not true in general.

By estimating the two quantities $m_\theta$ and $\Delta_\theta$, Angelopoulos et al. (2023b) derive an estimator and a CI for $\theta^\star$, which use both labelled and unlabelled data. The resulting CI is shorter than the classical confidence interval based solely on the labelled data when $N \gg n$ and $f$ is accurate because, in this case, $m_\theta$ can be estimated with low variance using the unlabelled data, while $\Delta_\theta$ is close to zero.

---

[1]Department of Statistics, University of Oxford. Correspondence to: Stefano Cortinovis <cortinovis@stats.ox.ac.uk>.

*Proceedings of the 42$^{nd}$ International Conference on Machine Learning*, Vancouver, Canada. PMLR 267, 2025. Copyright 2025 by the author(s).

Standard PPI employs off-the-shelf estimation and CI procedures for $\Delta_\theta$, which do not take advantage of any prior knowledge on the quality of the machine learning model $f$. However, in many applications, we expect the latter's predictions to be (i) usually very good, but (ii) sometimes prone to large errors and hallucinations. We propose to encode such an inductive bias with a horseshoe prior $\pi_\theta$ on $\Delta_\theta$ (Carvalho et al., 2010), which accommodates the aforementioned properties by exhibiting (i) an infinitely tall spike at the origin, and (ii) Cauchy-like tails at infinity. In order to construct valid confidence regions for $\Delta_\theta$ using the horseshoe prior, we resort to the frequentist-assisted by Bayes (FAB) framework (Pratt, 1961; 1963; Yu & Hoff, 2018). This approach provides confidence regions such that their expected length is lower for rectifiers $\Delta_\theta$ that have high probability under $\pi_\theta$, and larger otherwise. While the resulting confidence regions have exact coverage for any prior $\pi_\theta$, the horseshoe prior is particularly well-suited for PPI. Being concentrated around the origin, it produces shorter confidence regions when the predictions are good, i.e. $||\Delta_\theta|| \simeq 0$. At the same time, its heavy tails ensure robustness when the predictions are poor. Indeed, as shown by Cortinovis & Caron (2024), if $||\Delta_\theta|| \gg 0$, the FAB procedure with the horseshoe prior reverts to the traditional CI based on the sample mean.

In this work, we introduce FAB-PPI, a Bayes-assisted approach for PPI that encodes prior information on the quality of the machine learning predictions by specifying a prior for the rectifier $\Delta_\theta$. FAB-PPI is:

- Statistically valid, as its confidence regions have correct coverage for any choice of prior;

- Efficient, as its confidence regions have smaller expected length when the predictions are good;

- Robust, as it reverts to standard PPI when the predictions are poor, if the horseshoe prior is used;

- Modular, as it can be used in conjunction with power tuning (Angelopoulos et al., 2023b).

The remainder of the paper is organised as follows. Section 2 reviews related work. Section 3 provides background on control variates, PPI, and FAB confidence regions. Section 4 describes our novel approach for PPI, called FAB-PPI. Section 5 demonstrates the benefits of FAB-PPI on synthetic and real data. Finally, Section 6 discusses limitations and further extensions of our approach.

## 2. Related Work

PPI (Angelopoulos et al., 2023a) was introduced to obtain shorter CIs for the parameters of interest by leveraging machine learning predictions in semi-supervised settings. PPI has since been extended in multiple directions. PPI++ (Angelopoulos et al., 2023b) proposes a different, loss-based for-

mulation of PPI, leading to a more computationally efficient procedure, along with an additional power tuning parameter to enhance PPI's performance. Stratified PPI (Fisch et al., 2024) improves upon PPI by employing a data stratification strategy. Cross PPI (Zrnic & Candès, 2024b) demonstrates how the training of $f$ can be included in the PPI pipeline. Active statistical inference (Zrnic & Candès, 2024a) applies an active learning approach to select which inputs from the unlabelled set should be labelled. Closer to our work, Bayesian PPI (Hofer et al., 2024) considers an alternative PPI estimator motivated by Bayesian ideas. However, their approach provides Bayesian credible intervals, which do not offer frequentist guarantees. Additionally, their approach achieves similar experimental performance to PPI, while we demonstrate that FAB-PPI may significantly improve upon PPI.

As discussed in Angelopoulos et al. (2023a;b), PPI has close ties with control variates for variance reduction (Glasserman, 2003, §4.1). In the case of mean estimation, the form of the PPI estimator is similar to the one proposed by Zhang et al. (2019). PPI is also related to work in semiparametric inference with missing data (Robins & Rotnitzky, 1995).

The concept of Bayes-optimal confidence regions originates from the work of Pratt (1961; 1963). Pratt's approach, which has been given the name FAB by Yu & Hoff (2018), has since been extended in multiple directions (Brown et al., 1995; Farchione & Kabaila, 2008; Kabaila & Giri, 2013; Kabaila & Farchione, 2022; Yu & Hoff, 2018; Hoff & Yu, 2019; Hoff, 2023). In particular, Cortinovis & Caron (2024) show that, when combined with priors with power-law tails, FAB provides robust confidence regions that revert to classical ones in the presence of outliers. Hoff (2023) applied FAB in a predictive supervised context, showing that it can lead to more accurate predictions than standard methods.

## 3. Background

### 3.1. Control Variates

The method of control variates is a standard variance-reduction technique in Monte Carlo approximation (Glasserman, 2003, §4.1). For simplicity, we present the method in the scalar case, but extensions to the multivariate setting are available. Let $(Z, Y)$ be a pair of real-valued random variables, and assume we are interested in estimating $\mathbb{E}[Y]$ based on an iid sample $\{(Z_i, Y_i)\}_{i=1}^n$. Assuming $\mu = \mathbb{E}[Z]$ is known, one defines the control-variate estimator (CVE)

$$\widehat{Y}_\lambda^{\mathrm{cv}} = \overline{Y} - \lambda(\overline{Z} - \mu) = \frac{1}{n}\sum_{i=1}^n (Y_i - \lambda(Z_i - \mu)), \quad (5)$$

where $\lambda \in \mathbb{R}$ is a tuning coefficient and $\overline{Z}$ and $\overline{Y}$ are the sample means of $(Z_i)$ and $(Y_i)$, respectively. The centred random variable $Z_i - \mu$ serves as a control variate to estimate $\mathbb{E}[Y]$. The CVE is a consistent and

unbiased estimator of $\mathbb{E}[Y]$ with $\mathrm{var}(\widehat{Y}_\lambda^{\mathrm{cv}}) = (\mathrm{var}(Y) - 2\lambda\mathrm{cov}(Z,Y) + \lambda^2\mathrm{var}(Z))/n$, while $\mathrm{var}(\overline{Y}) = \mathrm{var}(Y)/n$. Therefore, the CVE has smaller variance than $\overline{Y}$ whenever $\lambda \in (\min\{0, 2\lambda^\star\}, \max\{0, 2\lambda^\star\})$, where the optimal coefficient is $\lambda^\star = \mathrm{cov}(Z,Y)/\mathrm{var}(Z)$. In this case, $\mathrm{var}(\widehat{Y}_{\lambda^\star}^{\mathrm{cv}}) = (1 - \rho_{Z,Y}^2)\mathrm{var}(\overline{Y})$, where $\rho_{Z,Y}$ is the correlation between $Z$ and $Y$. The more correlated $Z$ and $Y$, the larger the variance reduction. By plugging the estimator

$$\widehat{\lambda} = \frac{\sum_{i=1}^n (Z_i - \overline{Z})(Y_i - \overline{Y})}{\sum_{i=1}^n (Z_i - \overline{Z})^2} \tag{6}$$

for $\lambda$ in Equation (5), one has

$$\frac{\widehat{Y}_{\widehat{\lambda}}^{\mathrm{cv}} - \mathbb{E}[Y]}{s/\sqrt{n}} \to \mathcal{N}(0,1)$$

as $n \to \infty$, where $s$ is the sample standard deviation of $\{(Y_i - \widehat{\lambda}Z_i)\}_{i=1,\dots,n}$. Hence, $\widehat{Y}_{\widehat{\lambda}}^{\mathrm{cv}} \pm z_{1-\alpha/2}s/\sqrt{n}$ is an asymptotically valid $(1-\alpha)$ CI for $\mathbb{E}[Y]$, whose asymptotic width is $2z_{1-\alpha/2}\sqrt{1 - \rho_{Z,Y}^2}\sqrt{\mathrm{var}(Y)}/\sqrt{n}$.

### 3.2. Prediction-Powered Inference

PPI (Angelopoulos et al., 2023a) defines an estimator $\widehat{\theta}$ and a CI $\mathcal{C}_\alpha^{\mathrm{PP}}$ for a parameter of interest $\theta^\star$ satisfying Equation (2). In particular, let $\widehat{m}_\theta$ and $\widehat{\Delta}_\theta$ be some estimators of $m_\theta$ and $\Delta_\theta$. Using Equation (2), the estimator $\widehat{\theta}$ is defined as the solution, in $\theta$, to the equation

$$\widehat{m}_\theta + \widehat{\Delta}_\theta = 0. \tag{7}$$

Similarly, let $\mathcal{R}_\delta$ and $\mathcal{T}_{\alpha-\delta}$ be $1 - \delta$ and $1 - (\alpha - \delta)$ CIs for $\Delta_\theta$ and $m_\theta$, respectively. Then, the PPI confidence interval $\mathcal{C}_\alpha^{\mathrm{PP}}$ is defined as

$$\mathcal{C}_\alpha^{\mathrm{PP}} = \{\theta \mid 0 \in \mathcal{R}_\delta + \mathcal{T}_{\alpha-\delta}\}, \tag{8}$$

where $+$ denotes the Minkowski sum. Typical choices for $\widehat{m}_\theta$ and $\mathcal{T}_{\alpha-\delta}$ are the sample mean of the unlabelled data,

$$\widehat{m}_\theta = \frac{1}{N}\sum_{i=1}^N \mathcal{L}_\theta'(\widetilde{X}_i, f(\widetilde{X}_i)), \tag{9}$$

and classical CIs for sample means, respectively. Different choices for $\widehat{\Delta}_\theta$ have been proposed in the literature, leading to different PPI estimators.

**Standard PPI.** Angelopoulos et al. (2023a) propose to use the sample mean

$$\widehat{\Delta}_\theta^{\mathrm{PP}} = \frac{1}{n}\sum_{i=1}^n \left(\mathcal{L}_\theta'(X_i, Y_i) - \mathcal{L}_\theta'(X_i, f(X_i))\right) \tag{10}$$

as an estimator for $\Delta_\theta$ and the associated classical CIs to construct $\mathcal{R}_\delta$. For the squared loss, the estimator $\widehat{\theta}^{\mathrm{PP}}$ solving $\widehat{m}_\theta + \widehat{\Delta}_\theta = 0$ takes the control variate form

$$\widehat{\theta}^{\mathrm{PP}} = \overline{Y} - \left(\frac{1}{n}\sum_{i=1}^n f(X_i) - \frac{1}{N}\sum_{j=1}^N f(\widetilde{X}_j)\right) \tag{11}$$

with control variate $f(X_i) - \frac{1}{N}\sum_{j=1}^N f(\widetilde{X}_j)$ and $\lambda = 1$.

**PPI++.** Angelopoulos et al. (2023b) extend standard PPI by introducing an additional control-variate parameter $\lambda$, which they call power tuning parameter. The chosen $\widehat{m}_\theta$ is still the sample mean (9), while $\widehat{\Delta}_\theta^{\mathrm{PP+}}$ now takes the control variate form

$$\widehat{\Delta}_\theta^{\mathrm{PP+}} = \frac{1}{n}\sum_{i=1}^n \left(\mathcal{L}_\theta'(X_i, Y_i) - \mathcal{L}_\theta'(X_i, f(X_i))\right) \tag{12}$$
$$- (\widehat{\lambda} - 1)\left(\frac{1}{n}\left[\sum_{i=1}^n \mathcal{L}_\theta'(X_i, f(X_i))\right] - \widehat{m}_\theta\right),$$

where $\widehat{\lambda}$ is estimated from the data. In this case, the centred control variate is $\mathcal{L}_\theta'(X_i, f(X_i)) - \widehat{m}_\theta$, which depends only on the machine learning predictions. For the squared loss, we obtain

$$\widehat{\theta}^{\mathrm{PP+}} = \overline{Y} - \widehat{\lambda}\left(\frac{1}{n}\sum_{i=1}^n f(X_i) - \frac{1}{N}\sum_{j=1}^N f(\widetilde{X}_j)\right) \tag{13}$$

with plug-in estimator

$$\widehat{\lambda} = \frac{c_n}{(1 + \frac{n}{N})v_{n+N}}, \tag{14}$$

where $c_n$ is the sample covariance of $(Y_i, f(X_i))_{i=1}^n$ and $v_{n+N}$ is the sample variance of $((f(X_i))_{i=1}^n, (f(\widetilde{X}_j))_{j=1}^N)$. The estimator (13) is closely related (though slightly different) to the one introduced by Zhang et al. (2019) for mean estimation in semi-supervised inference.

**CLT-based CIs.** While the definition of the PPI confidence interval (8) allows for merging any CIs $\mathcal{R}_\delta$ and $\mathcal{T}_{\alpha-\delta}$ for $\Delta_\theta$ and $m_\theta$, in practice these are often chosen to be CLT-based CIs that, once combined into $\mathcal{C}_\alpha^{\mathrm{PP}}$ give exact asymptotic coverage,

$$\liminf_{n,N\to\infty} \mathrm{Pr}(\theta^\star \in \mathcal{C}_\alpha^{\mathrm{PP}}) \geq 1 - \alpha.$$

Such CLT-based CIs rely on the following standard assumption on the estimators $\widehat{m}_\theta$ and $\widehat{\Delta}_\theta$.

**Assumption 3.1** (CLT assumption for PPI and PPI++). Let $\widehat{m}_\theta$ be the sample mean (9) and consider some estimator $(\widehat{\sigma}_\theta^f)^2$ of $\mathrm{var}(\widehat{m}_\theta)$, with $(\widehat{\sigma}_\theta^f)^2/\mathrm{var}(\widehat{m}_\theta) \to 1$ almost surely.

Let $\widehat{\Delta}_\theta$ be either the PPI estimator (10) or the PPI++ estimator (12) and consider some estimator $\widehat{\sigma}_\theta$ of $\text{var}(\widehat{\Delta}_\theta)$ with $\widehat{\sigma}_\theta/\text{var}(\widehat{\Delta}_\theta) \to 1$ a.s. Assume that, as $\min(n, N) \to \infty$,

$$(\widehat{m}_\theta - m_\theta)/\widehat{\sigma}_\theta^f \to \mathcal{N}(0, 1) \tag{15}$$

$$(\widehat{\Delta}_\theta - \Delta_\theta)/\widehat{\sigma}_\theta \to \mathcal{N}(0, 1). \tag{16}$$

### 3.3. Bayes-Optimal Confidence Regions

The FAB framework (Pratt, 1961; 1963; Yu & Hoff, 2018) aims to construct valid confidence regions with smaller expected volume. Let $W \mid \beta \sim \mathcal{N}(\beta, \sigma^2)$ with some prior $\pi_0(\beta)$ and denote by $\pi(w) = \int p(w \mid \beta)\pi_0(\beta)d\beta$ the corresponding marginal likelihood. For $\alpha \in (0, 1)$, let $\mathcal{C}_\alpha(w)$ be an exact $(1 - \alpha)$ confidence region for $\beta$ based on the data $w$. That is, for any fixed $\beta_0$,

$$\Pr(\beta \in \mathcal{C}_\alpha(W) \mid \beta = \beta_0) = 1 - \alpha. \tag{17}$$

Let $\text{vol}(\mathcal{C}_\alpha(w)) = \int_{\beta' \in \mathcal{C}_\alpha(w)} d\beta'$ be the volume of $\mathcal{C}_\alpha(w)$, and consider its expected value under the marginal likelihood $\pi$,

$$\mathbb{E}[\text{vol}(\mathcal{C}_\alpha(W))] = \int \text{vol}(\mathcal{C}_\alpha(w))\pi(w)dw. \tag{18}$$

**Definition 3.2.** For $\alpha \in (0, 1)$, $\sigma > 0$ and a prior $\pi_0(\beta)$, the FAB confidence region $\mathcal{C}_\alpha$ for the mean parameter $\beta$ of the normal model $Y \mid \beta \sim \mathcal{N}(\beta, \sigma^2)$, is the minimiser of the (Bayesian) expected volume

$$\mathcal{C}_\alpha = \arg\min_{\widetilde{\mathcal{C}}_\alpha} \mathbb{E}[\text{vol}(\widetilde{\mathcal{C}}_\alpha(W))] \tag{19}$$

subject to the (frequentist) coverage constraint (17). We write $\mathcal{C}_\alpha(w) = \text{FAB-CR}(w; \pi_0, \sigma^2, \alpha)$.

The solution to Equation (19), which exists and is unique if $\pi_0(\beta)$ is not degenerate (Cortinovis & Caron, 2024, Theorem 3.3), may be found numerically as long as the marginal likelihood $\pi(w)$ can be evaluated pointwise. Additional details are provided in Section S1.2. Intuitively, the FAB confidence region $\mathcal{C}_\alpha(w)$ constructed through Equation (19) will be smaller for values of $w$ that are likely under the marginal likelihood, and larger otherwise. As a result of this, while FAB guarantees the right coverage for any prior, one that assigns high probability to the value of $\beta$ that generated the data is required to achieve smaller expected volume compared to the standard CI ($w \pm \sigma z_{1-\alpha/2}$), whose width does not depend on $w$.

**Bayes-Assisted Estimator.** A natural estimator to use alongside the FAB confidence region $\mathcal{C}_\alpha(w)$ is the posterior mean $\widehat{\beta}(W) = \mathbb{E}[\beta \mid W]$. As shown by (Cortinovis & Caron, 2024, Theorem 3.3), it is always contained within the confidence region: $\widehat{\beta}(w) \in \mathcal{C}_\alpha(w)$ for any $w \in \mathbb{R}$ and any $\alpha \in (0, 1)$. We refer to $\widehat{\beta}(W)$ as the Bayes-assisted estimator.

## 4. FAB-PPI

Our approach, which we call FAB-PPI, combines the PPI framework with the FAB construction of confidence regions by specifying a prior on the rectifier $\Delta_\theta$. To ease the presentation, here we describe the method for $Y, \theta \in \mathbb{R}$. The general multivariate case is discussed in Section S4.

As in PPI, we use the sample mean (9) as the estimator of $m_\theta$. For $\Delta_\theta$, we start by considering a consistent estimator $\widehat{\Delta}_\theta$, such as the sample mean (10) used in PPI, or the control variate estimator (12) used in PPI++. Throughout this section, we assume that Assumption 3.1 is satisfied. That is, a CLT holds for $\widehat{m}_\theta$ and $\widehat{\Delta}_\theta$ with respect to some estimators $(\widehat{\sigma}_\theta^f)^2$ and $\widehat{\sigma}_\theta^2$ of $\text{var}(\widehat{m}_\theta)$ and $\text{var}(\widehat{\Delta}_\theta)$, respectively. In this setting, let $\pi_0(\Delta_\theta; \tau_n)$ be a prior on $\Delta_\theta$ with scale parameter $\tau_n$, which may depend on the labelled data through $\widehat{\sigma}_\theta$. Denote by $\ell(w; \sigma, \tau)$ the log-marginal likelihood, evaluated at $w$, of a Gaussian likelihood model with mean $\Delta$ and variance $\sigma^2$ under the prior $\pi_0(\Delta; \tau)$,

$$\ell(w; \sigma, \tau) = \log \int_{\mathbb{R}} \mathcal{N}(w; \Delta, \sigma^2)\pi_0(\Delta; \tau)d\Delta.$$

### 4.1. Bayes-Assisted PPI Estimators

Consider the Bayes-assisted estimator

$$\widehat{\Delta}_\theta^{\text{FABPP}} = \widehat{\Delta}_\theta + \widehat{\sigma}_\theta^2 \ell'\left(\widehat{\Delta}_\theta; \widehat{\sigma}_\theta, \tau_n\right) \tag{20}$$

for the rectifier $\Delta_\theta$. By Tweedie's formula (Efron, 2011), the above estimator is the posterior mean of the mean parameter of a Gaussian likelihood model under the prior $\pi_0$. Note however that we do not assume here that $\widehat{\Delta}_\theta$ is normally distributed for a fixed $n$.

The FAB-PPI estimator of $\theta^\star$, denoted by $\widehat{\theta}^{\text{FABPP}}$, is then obtained as the solution, in $\theta$, to the equation

$$\widehat{m}_\theta + \widehat{\Delta}_\theta^{\text{FABPP}} = 0.$$

### 4.2. FAB-PPI Confidence Regions

As in PPI, let $\mathcal{T}_{\alpha-\delta}(\widehat{m}_\theta)$ denote a standard $1 - (\alpha - \delta)$ confidence interval for $m_\theta$. For $\Delta_\theta$, we apply the FAB framework with the prior $\pi_0$ to obtain a $1 - \delta$ confidence region $\mathcal{R}_\delta^{\text{FABPP}}(\widehat{\Delta}_\theta) = \text{FAB-CR}(\widehat{\Delta}_\theta; \pi_0(\cdot; \tau_n), \widehat{\sigma}_\theta, \delta)$. Then, the FAB-PPI confidence region $\mathcal{C}_\alpha^{\text{FABPP}}$ is obtained as

$$\mathcal{C}_\alpha^{\text{FABPP}} = \left\{\theta \mid 0 \in \mathcal{R}_\delta^{\text{FABPP}}(\widehat{\Delta}_\theta) + \mathcal{T}_{\alpha-\delta}(\widehat{m}_\theta)\right\}. \tag{21}$$

Algorithm 1 summarises the steps of the FAB-PPI approach in a general convex estimation problem.

### 4.3. Choosing the Prior

FAB-PPI is motivated by applications in which the PPI predictor $f$ is expected to be generally accurate, as measured

**Algorithm 1** FAB-PPI for convex estimation

**Input:** labelled $\{(X_i, Y_i)\}_{i=1}^n$, unlabelled $\{\widetilde{X}_j\}_{j=1}^N$, predictor $f$, prior $\pi_0(\cdot\;; \tau_n)$, error levels $\alpha, \delta$

    Set $\widehat{\lambda} = 1$ (FAB-PPI) or estimate $\widehat{\lambda}$ from data (FAB-PPI++) as in Angelopoulos et al. (2023b).

    **for** $\theta \in \Theta_{\text{grid}}$ **do**

$$\widehat{m}_\theta \leftarrow \frac{1}{N}\sum_{i=1}^N \mathcal{L}'_\theta(\widetilde{X}_i, f(\widetilde{X}_i))$$

$$\widehat{\xi} \leftarrow \frac{1}{n}\sum_{i=1}^n \left(\mathcal{L}'_\theta(X_i, Y_i) - \widehat{\lambda}\mathcal{L}'_\theta(X_i, f(X_i))\right)$$

$$\widehat{\Delta}_\theta \leftarrow \widehat{\xi} + (\widehat{\lambda} - 1)\widehat{m}_\theta$$

$$\widehat{\sigma}_m^2 \leftarrow \frac{1}{N-1}\sum_{i=1}^N \left(\mathcal{L}'_\theta(\widetilde{X}_i, f(\widetilde{X}_i)) - \widehat{m}_\theta\right)^2$$

$$\widehat{\sigma}_\xi^2 \leftarrow \frac{1}{n-1}\sum_{i=1}^n \left(\mathcal{L}'_\theta(X_i, Y_i) - \widehat{\lambda}\mathcal{L}'_\theta(X_i, f(X_i)) - \widehat{\xi}\right)^2$$

$$\widehat{\sigma}_\theta^2 \leftarrow \frac{1}{n}\widehat{\sigma}_\xi^2 + \frac{(\widehat{\lambda}-1)^2}{N}\widehat{\sigma}_m^2$$

$$\mathcal{T}_{\alpha-\delta}(\widehat{m}_\theta) \leftarrow \left(\widehat{m}_\theta \pm \frac{\widehat{\sigma}_m}{\sqrt{N}}z_{1-(\alpha-\delta)/2}\right)$$

$$\mathcal{R}_\delta^{\text{FABPP}}(\widehat{\Delta}_\theta) \leftarrow \text{FAB-CR}(\widehat{\Delta}_\theta; \pi_0(\cdot\;;\tau_n), \widehat{\sigma}_\theta, \delta)$$

$$\widehat{\Delta}_\theta^{\text{FABPP}} \leftarrow \widehat{\Delta}_\theta + \widehat{\sigma}_\theta^2 \ell'\left(\widehat{\Delta}_\theta; \widehat{\sigma}_\theta, \tau_n\right)$$

    **end for**

**Outputs:** estimator $\widehat{\theta}^{\text{FABPP}} = \arg\min_{\Theta_{\text{grid}}}\left|\widehat{m}_\theta + \widehat{\Delta}_\theta^{\text{FABPP}}\right|$ and CR $\mathcal{C}_\alpha^{\text{FABPP}} = \left\{\theta \mid 0 \in \mathcal{R}_\delta^{\text{FABPP}}(\widehat{\Delta}_\theta) + \mathcal{T}_{\alpha-\delta}(\widehat{m}_\theta)\right\}$

---

by the rectifier $\Delta_\theta$. Such a property may be encoded in $\pi_0(\Delta_\theta; \tau_n)$ by choosing a prior that concentrates around zero. As mentioned in Section 3.3, the FAB construction of $\mathcal{R}_\delta^{\text{FABPP}}(\widehat{\Delta}_\theta)$ will exhibit smaller volume compared to the classical CI, and hence result in downstream efficiency gains over standard PPI, if the true rectifier $\Delta_\theta$ is likely under $\pi_0$. In particular, the prior scale $\tau_n$ controls the size of the potential efficiency gains and losses of FAB-PPI over PPI: the smaller $\tau_n$, the more the resulting CR will shrink (resp. grow) when $\Delta_\theta \simeq 0$ (resp. $|\Delta_\theta| \gg 0$). Experimentally, we find that the choice $\tau_n = \widehat{\sigma}_\theta$ results in a parameter-free approach that strikes a good compromise. More general choices of $\tau_n$ are briefly mentioned in Section 6.

A seemingly natural proposal for $\pi_0$ that meets the requirements above is the Gaussian prior

$$\pi_N(\Delta_\theta; \widehat{\sigma}_\theta) = \mathcal{N}(\Delta_\theta; 0, \widehat{\sigma}_\theta). \tag{22}$$

However, as we will discuss in Section 4.4, $\pi_N$ exhibits undesirable properties for FAB-PPI. Instead, we propose to use the horseshoe prior (Carvalho et al., 2010)

$$\pi_{\text{HS}}(\Delta_\theta; \widehat{\sigma}_\theta) = \int_0^\infty \mathcal{N}(\Delta_\theta; 0, \nu^2\widehat{\sigma}_\theta^2)C^+(\nu; 0, 1)d\nu, \tag{23}$$

where $C^+(\nu; 0, 1)$ denotes the pdf of the half-Cauchy distribution with location parameter 0 and scale parameter 1. In the case of $\pi_{\text{HS}}$, the choice of scaling $\tau_n = \widehat{\sigma}_\theta$ is further

motivated by Piironen & Vehtari (2017, §3.3). Furthermore, the horseshoe prior has power-law tails, making it a particularly robust choice for FAB-PPI, as discussed in Section 4.4. Crucially, for both priors $\pi_N$ and $\pi_{\text{HS}}$, the marginal likelihood under a Gaussian model with standard deviation $\widehat{\sigma}_\theta$ can be expressed in terms of standard functions (see Section S1.1 for the horseshoe), enabling us to compute $\widehat{\Delta}_\theta^{\text{FABPP}}$ and $\mathcal{R}_\delta^{\text{FABPP}}(\widehat{\Delta}_\theta)$ in Algorithm 1.

### 4.4. Theoretical Properties

As shown by the following result, proved in Section S3.1, the FAB-PPI CR has exact asymptotic coverage.

**Theorem 4.1** (Asymptotic coverage). *For $\alpha \in (0, 1)$, let $\mathcal{C}_\alpha^{\text{FABPP}}$ be the FAB-PPI confidence region* (21) *under the Gaussian prior* (22) *or the horseshoe prior* (23). *Then, under Assumption 3.1,*

$$\liminf_{\min(n,N)\to\infty} \Pr(\theta^\star \in \mathcal{C}_\alpha^{\text{FABPP}}) \geq 1 - \alpha.$$

The proof of Theorem 4.1 crucially relies on showing exact asymptotic coverage of the FAB CR $\mathcal{R}_\delta^{\text{FABPP}}(\widehat{\Delta}_\theta)$. While the latter holds for both priors introduced in the previous sections, the two limits behave very differently. In particular, as discussed in Remark S3.4, the volume of $\mathcal{R}_\delta^{\text{FABPP}}(\widehat{\Delta}_\theta)$ vanishes asymptotically under $\pi_{\text{HS}}$, while it does not under $\pi_N$.

The behaviour of $\mathcal{R}_\delta^{\text{FABPP}}(\widehat{\Delta}_\theta)$ under the two priors also differs for large values of observed $\widehat{\Delta}_\theta$. In case of increasing disagreement between the prior and the data, Gaussian FAB confidence regions are known to become arbitrarily large (Yu & Hoff, 2018). On the other hand, thanks to its power-law tails, the horseshoe results in confidence regions that revert to the corresponding standard CI (Cortinovis & Caron, 2024). Here, we state the implication of this property on FAB-PPI informally, and provide a formal proof in Section S3.2.

**Proposition 4.2** (Robustness under the horseshoe, informal). *For $\alpha \in (0, 1)$, let $\mathcal{C}_\alpha^{\text{FABPP}}$ and $\mathcal{C}_\alpha^{\text{PP}}$ denote, respectively, the FAB-PPI confidence region* (21) *under the horseshoe prior* (23) *and the standard CLT-based PPI CI for $\theta$, both viewed as functions of $\widehat{\Delta}_\theta$. If $|\widehat{\Delta}_\theta| \gg 0$, then*

$$\mathcal{C}_\alpha^{\text{FABPP}} \simeq \mathcal{C}_\alpha^{\text{PP}}.$$

In practice, this means that, in the presence of heavily biased predictors, FAB-PPI with the horseshoe prior reverts to standard PPI. In a sense, this represents a form of robustness to prior misspecification of FAB-PPI under the horseshoe.

Overall, Remark S3.4 and Proposition 4.2 provide strong support for preferring $\pi_{\text{HS}}$ over $\pi_N$ within the FAB-PPI framework.

## 4.5. FAB-PPI for Mean Estimation

To provide a concrete example, a specialised version of Algorithm 1 under the squared loss is derived in Section S2.1. Here, we briefly discuss the differences between the FAB-PPI mean estimator and its standard PPI counterpart, as well as the asymptotic behaviour of the former. Under the squared loss, the rectifier $\Delta := \Delta_\theta$ does not depend on $\theta$ and the FAB-PPI estimator $\widehat{\theta}^{\mathrm{FABPP}}$ corresponding to the chosen estimator $\widehat{\Delta}$ (PPI or PPI++) is given by

$$\widehat{\theta}^{\mathrm{FABPP}} = \widehat{\theta} - \widehat{\sigma}^2 \ell'(\widehat{\Delta}; \widehat{\sigma}, \tau_n) \qquad (24)$$
$$= \overline{Y} - \widehat{\lambda}\left(\frac{1}{n}\sum_{i=1}^{n} f(X_i) - \frac{1}{N}\sum_{j=1}^{N} f(\widetilde{X}_j)\right)$$
$$- \widehat{\sigma}^2 \ell'\left(\widehat{\Delta}; \widehat{\sigma}, \tau_n\right),$$

where $\widehat{\sigma}^2$ is an estimator of $\mathrm{var}(\widehat{\Delta})$, $\widehat{\theta}$ is the PPI estimator corresponding to $\widehat{\Delta}$, and $\widehat{\lambda}$ is set either to one (PPI) or (14) (PPI++). In both cases, the estimator $\widehat{\theta}^{\mathrm{FABPP}}$ takes the form

Classic Estimator + PPI correction + **Bayes correction**,

where the last component depends on the chosen prior. The following proposition, proved in Section S3.3, further differentiates between the priors presented in Section 4.3 in favour of the horseshoe.

**Proposition 4.3** (Consistency of FAB-PPI mean estimators).
*Let $\widehat{\theta}_{HS}^{FABPP}$ and $\widehat{\theta}_{N}^{FABPP}$ be the FAB-PPI estimators (24) under the horseshoe (23) and Gaussian (22) priors, respectively. If the PPI estimator $\widehat{\theta}$ is a consistent estimator of $\theta^\star$, then $\widehat{\theta}_{HS}^{FABPP}$ is a consistent estimator of $\theta^\star$, while $\widehat{\theta}_{N}^{FABPP}$ is not.*

Intuitively, this is due to the fact that the influence of $\pi_{\mathrm{HS}}$ vanishes asymptotically, while for $\pi_{\mathrm{N}}$ it does not.

# 5. Experiments

We compare FAB-PPI and power-tuned FAB-PPI (FAB-PPI++) to classical inference, PPI and power-tuned PPI (PPI++) on both synthetic and real estimation problems. For FAB-PPI, we use (HS) and (N) to indicate the use of the horseshoe and Gaussian priors defined in Section 4.3. As already mentioned, PPI is motivated by settings in which labelled data are scarce, while unlabelled data are abundant. Moreover, the application of FAB to PPI specifically targets the estimation of the rectifier $\Delta_\theta$. For these reasons, we choose to focus on cases where $N \gg n$ is large enough to rule out any uncertainty in the measure of fit $m_\theta$, which we estimate using the sample mean $\widehat{m}_\theta$ (9). As a result of this, given a $1 - \delta$ confidence interval (FAB or not) $\mathcal{R}_\delta$ for $\Delta_\theta$, the corresponding $1 - \alpha$ CI for $\theta^\star$ is obtained simply by setting $\delta = \alpha$ and shifting $\mathcal{R}_\delta$ by $\widehat{m}_\theta$. This simplification allows us to evaluate the direct effect of FAB on the procedure,

eliminating concerns about the loss of tightness in the CI on $\theta^\star$ due to the Minkowski sum in Equation (21). In all experiments, we check empirically that $N$ is large enough to make this assumption by monitoring the coverage of the resulting intervals against both the nominal level $1 - \alpha$ and the coverage of PPI intervals that also consider the uncertainty in $m_\theta$ (denoted with PPI (full) and PPI++ (full) in Section S6).

## 5.1. Synthetic Data

The simulated experiments below have a common structure. We sample two datasets, $n$ labelled observations $\{(X_i, Y_i)\}_{i=1}^{n}$ iid from $\mathbb{P}$ and $N$ unlabelled observations $\{\widetilde{X}_i\}_{i=1}^{N}$ iid from $\mathbb{P}_X$. We use a prediction rule $f$ to obtain predictions $\{f(X_i)\}_{i=1}^{n}$ and $\{f(\widetilde{X}_i)\}_{i=1}^{N}$. We apply the different procedures to obtain estimates and $1 - \alpha$ confidence regions for the mean $\theta^\star = \mathbb{E}[Y]$. For all experiments, we set $\alpha = 0.1$ and report the average mean squared error (MSE), interval volume, and coverage over 1000 repetitions.

**Biased Predictions.** We sample $X_i \overset{iid}{\sim} \mathcal{N}(0, 1)$ and $Y_i = X_i + \epsilon_i$ with $\epsilon_i \overset{iid}{\sim} \mathcal{N}(0, 1)$, so that $\theta^\star = \mathbb{E}[Y] = 0$. The prediction rule is defined as $f(X_i) = X_i + \gamma$, where $\gamma \in \mathbb{R}$. For this choice, the bias of $f$ is controlled by $\gamma$, since $\mathrm{MSE}(f) = \gamma^2 + 1$. For this experiment, we assume that $N$ is infinite, set $n = 200$, and vary $\gamma$ between $-1.5$ and $1.5$. Figure 1 shows the average interval volume as a function of $\gamma$ for classical inference, PPI++, and FAB-PPI++ with both a horseshoe and a Gaussian prior. Results for the non power-tuned meth-

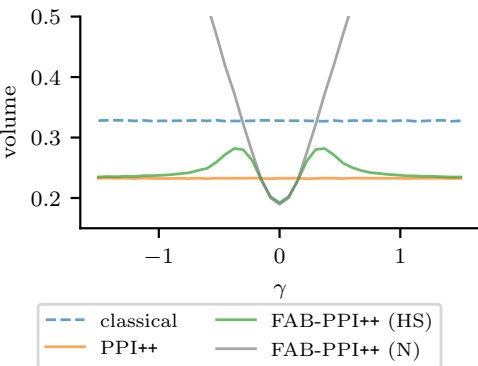

*Figure 1.* Biased predictions study. The panel shows the average CI volume as the bias level $\gamma$ varies.

ods, as well as MSE and coverage plots, are reported in Figure S6. Except for the version with the Gaussian prior, all the PPI procedures outperform classical inference for every bias level $\gamma$, but the behaviour exhibited by PPI++ deserves attention, as its CI volume is approximately constant across values of $\gamma$. This is due to the fact that, since $N$ is taken to be infinite and $n$ is fairly large, $\widehat{\lambda} \simeq \mathrm{cov}(Y, f(X)) = 1$ and the rectifier is accurately estimated with similar variance

across all values of $\gamma$. On the other hand, the CI volume for the FAB-PPI methods varies greatly with $\gamma$. When the bias is small ($\gamma \simeq 0$), the observed rectifier has a value close to 0, leading to smaller CIs. As the bias increases, the volume of the confidence intervals grows, until it surpasses that of the PPI intervals. At this point, the two FAB-PPI procedures behave differently: the volume of the Gaussian intervals grows without bound, whereas the horseshoe intervals eventually revert to the PPI ones. This example clearly shows that FAB-PPI with a horseshoe prior allows to obtain smaller CIs when the predictions are good, while ensuring robustness as the quality of the predictions decreases (Proposition 4.2).

**Noisy Predictions.** We consider the mean estimation example of Angelopoulos et al. (2023b, §7.1.1), which does not involve any covariate $X$. We sample $Y_i \stackrel{iid}{\sim} \mathcal{N}(0,1)$, so that $\theta^\star = \mathbb{E}[Y] = 0$. The prediction rule is defined as $f(X_i) = Y_i + \sigma_Y \epsilon_i$, where $\epsilon_i \stackrel{iid}{\sim} \mathcal{N}(0,1)$ and $\sigma_Y$ is successively set to 0.1, 1, and 2. For this experiment, we set $N = 10^6$ and vary $n$ from 100 to 1000. Figure 2 shows the average interval volume as a function of $n$ for the different methods as the noise level $\sigma_Y$ varies, while similar plots for the MSE and coverage are reported in Figure S7. In this

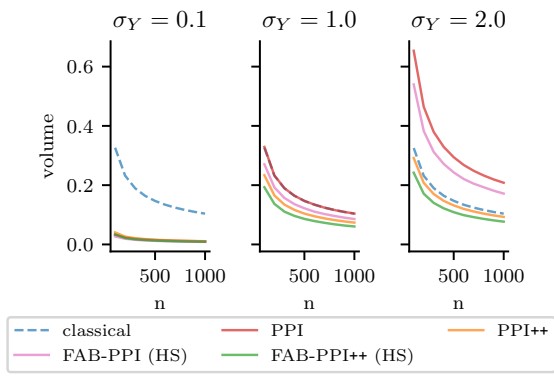

*Figure 2.* Noisy predictions study. The left, middle and right panels show the average CI volume for noise levels $\sigma_Y = 0.1, 1, 2$.

case, the effect of power tuning matches the observations of Angelopoulos et al. (2023b): as the noise level increases, $\hat{\lambda}$ decreases and less weight is given to the predicted labels. When the noise is small, all PPI procedures perform similarly, and much better than classical inference. When the noise is large, the power-tuned procedures perform similarly to or better than classical inference, whereas the non-tuned alternatives lose ground. At the intermediate noise level, the power-tuned methods clearly outperform the other baselines. Crucially, FAB-PPI outperforms the PPI counterpart at all noise levels, with FAB-PPI++ being the best performer overall. This is because, in this setting, while predictions exhibit increasing variance with $\sigma_Y$, they remain unbiased.

As a result of this, regardless of the value of $\lambda$ used, any additional shrinkage performed on the rectifier by FAB-PPI is beneficial. This example shows that FAB-PPI++ retains the benefits of power tuning, while also taking advantage of the adaptive shrinkage provided by the FAB procedure.

### 5.2. Real Data

We consider several estimation experiments using the datasets presented in Angelopoulos et al. (2023a) and briefly described in Section S5.1. Each dataset comes with covariate/label/prediction triples $\{X_i, Y_i, f(X_i)\}_{i=1}^N$, which we randomly split into two subsets with $n$ labelled and $N - n$ unlabelled observations, for varying values of $n$. For all experiments and methods, we report the average estimation MSE, CI volume and coverage across multiple repetitions.

We begin with four experiments, where the machine learning predictions provided are of high quality, and whose goals are as follows. Two of them are mean estimation tasks performed on the GALAXIES and FOREST datasets. The third one, performed on the ALPHAFOLD dataset, is an odds ratio estimation task, for which the construction of confidence intervals also indirectly involves mean estimation as detailed in Section S5.1. The fourth one, involving the HEALTHCARE dataset, is a logistic regression task. Figure 3 shows the results for classical inference, PPI++, and FAB-PPI++ applied to the datasets involving mean estimation.

The mean estimation results for the non power-tuned methods are reported in Figure S8, whereas the ones for the logistic regression experiment are reported in Figure S11. In all cases, FAB-PPI/FAB-PPI++ outperform classical inference and the corresponding PPI methods, both in terms of MSE and CI volume, while achieving comparable coverage. These examples suggest that the quality of the predictions of existing machine learning models on several real datasets may fall into the regime where the adaptive shrinkage provided by the FAB framework leads to a further improvement over standard PPI. In these settings, as the predictions are good, FAB-PPI under the horseshoe and Gaussian priors exhibit similar gains, as already seen in Figure 1.

However, the same is not true in the presence of bad predictions. For instance, Figure S12 shows the results of a quantile estimation experiment on the GENES dataset, where predictions are heavily biased. In this case, the behaviour of the FAB-PPI methods under the horseshoe and Gaussian priors differs significantly: the former matches the performance of the PPI methods, which outperform classical inference, whereas the latter leads to much larger MSE and CIs. As previously discussed, such desirable behaviour of FAB-PPI under the horseshoe prior is due to its robustness against large bias levels (Proposition 4.2). Similarly, Figure S13 reports the results of a linear regression experiment on the CENSUS dataset. For one of the two parameters con-

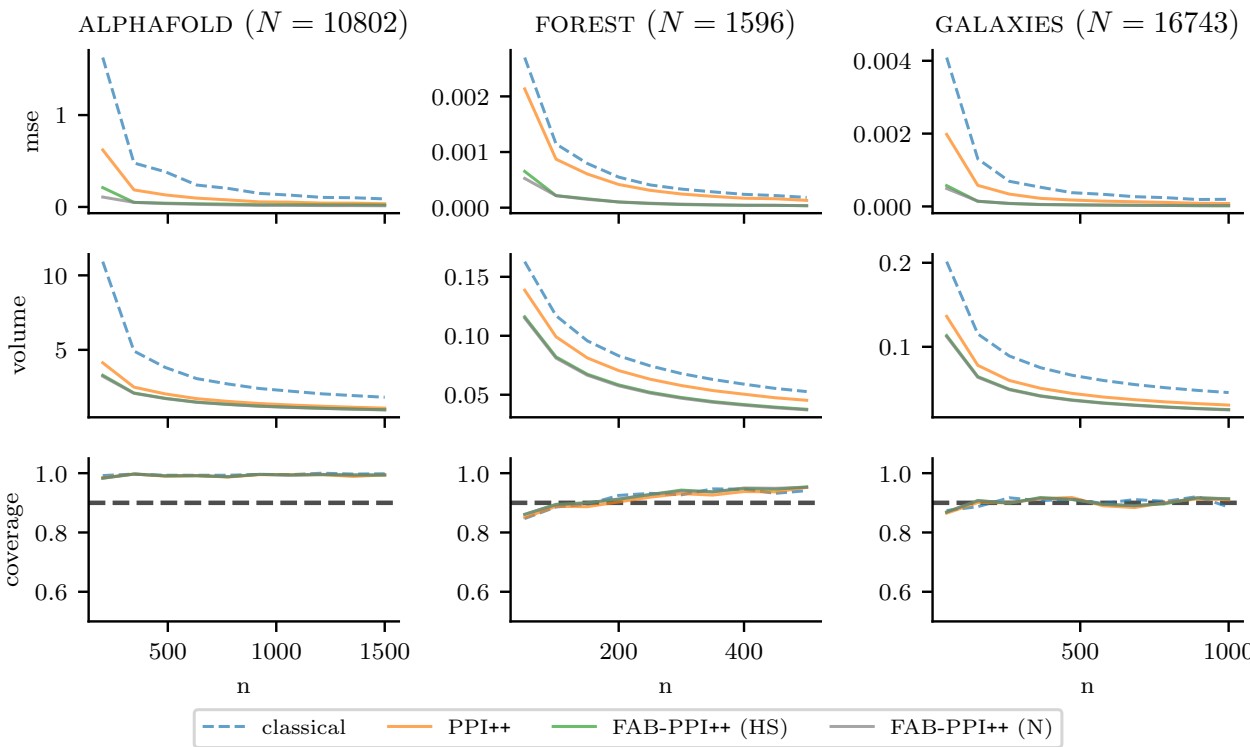

*Figure 3.* Real data mean estimation study. The left, middle, and right panels correspond to the ALPHAFOLD, GALAXIES, and FOREST datasets. The top, middle, and bottom rows show average MSE, CI volume, and CI coverage over 1000 repetitions for $\alpha = 0.1$.

sidered (panel (a)), FAB-PPI underperforms the alternatives under both priors for small $n$. However, as $n$ increases, the performance under the horseshoe prior improves and eventually matches that of the PPI methods, while the Gaussian prior does not. This example shows another facet of the horseshoe's robustness: even for moderate bias levels, as the available labelled sample size grows, disagreements between the prior and the data become apparent (i.e. $\mathrm{var}(\widehat{\Delta}_\theta)$ decreases), eventually leading Proposition 4.2 to take effect.

## 6. Discussion and Extensions

We proposed FAB-PPI as a Bayes-informed method to significantly improve the performance of PPI in the presence of high-quality predictions. In doing so, we showed that the horseshoe represents a sensible default prior for FAB-PPI, contrary to the seemingly natural choice of a Gaussian prior. However, several options may be worth exploring.

In particular, the horseshoe prior was chosen due to its popularity and key properties: (i) its spike at zero (ii) power-law tails and (iii) the closed-form expression for the marginal density $\pi(y)$. However, many other scale-mixture of Gaus-

sians models share these properties. For example, the family of priors with a beta prime (aka inverted beta) prior over the variance (Polson & Scott, 2012), which includes the horseshoe, normal-exponential-gamma (Griffin & Brown, 2011) and other robust priors (Berger, 1980; Strawderman, 1971) as special cases, shares the same three properties. On the other hand, some other standard priors such as the Laplace prior (Park & Casella, 2008) or normal-gamma prior (Caron & Doucet, 2008; Griffin & Brown, 2010) do not have power-law tails and therefore do not offer the same robustness guarantees. Other priors, such as the Student-t, lack an analytical expression for $\pi(y)$, therefore requiring additional numerical approximation to be applied to FAB-PPI.

Furthermore, we used the scale $\sigma$ of the noise in the generative model as the scale for both the horseshoe and Gaussian priors, as this allows us to obtain a simple, parameter-free approach, which generally performs well. Alternatively, one could consider a prior scale of $\eta\sigma$, where $\eta$ is a hyperparameter to be tuned using a validation set. However, in the case of the horseshoe, this renders the marginal likelihood intractable. While using a rescaled horseshoe prior for FAB-PPI remains feasible through numerical integration, as shown in Figure S10, this increases the

computational cost of the method. By contrast, a rescaled Gaussian prior would not encounter this issue. Furthermore, we conjecture that choosing a scale that does not depend on $\sigma$ may resolve the inconsistency of the estimator based on a Gaussian prior, which was discussed in Section 4.5.

As a potential drawback, FAB-PPI shares the computational limitations of the PPI approach (Angelopoulos et al., 2023a), which are discussed in Angelopoulos et al. (2023b). In particular, except for special cases such as mean estimation and linear regression, the method requires evaluating $\widehat{m}_\theta + \widehat{\Delta}_\theta$ over a grid of values of $\theta$. This can be computationally expensive, especially in high-dimensional settings.

**Supplementary Material and Code.** The supplementary material contains additional background, proofs, and experiments. All sections, figures, and equations in the supplementary material are prefixed with 'S' for clarity. Code for reproducing the experiments is available at https://github.com/stefanocortinovis/fab-ppi.

## Acknowledgements

Stefano Cortinovis is supported by the EPSRC Centre for Doctoral Training in Modern Statistics and Statistical Machine Learning (EP/S023151/1).

## Impact Statement

This paper presents work whose goal is to advance the field of Machine Learning. There are many potential societal consequences of our work, none which we feel must be specifically highlighted here.

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

## S1. Additional Background Material

### S1.1. Horseshoe Prior

Consider the Gaussian likelihood model

$$Y \mid \beta \sim \mathcal{N}(\beta, \sigma^2)$$

with standard deviation $\sigma > 0$ and mean parameter $\beta \in \mathbb{R}$. The horseshoe prior (Carvalho et al., 2010) with density $\pi_{\mathrm{HS}}$ can be represented as a scale mixture of normals (Andrews & Mallows, 1974)

$$\beta \mid \nu^2 \sim \mathcal{N}(0, \eta^2 \sigma^2 \nu^2) \tag{S25}$$
$$\nu \sim C^+(0, 1), \tag{S26}$$

where $\eta > 0$ and $C^+(0, 1)$ is the half-Cauchy distribution with location parameter 0 and scale parameter 1. Throughout this section and the main text, we assume $\eta = 1$. The rationale for this choice, along with a discussion of the general case $\eta \neq 1$, is provided at the end of this section.

The marginal likelihood is given by

$$
\begin{aligned}
\pi(y) &= \int_{-\infty}^{\infty} \mathcal{N}(y \mid \beta, \sigma^2) \pi_{\mathrm{HS}}(\beta) d\beta \\
&= \frac{1}{\sqrt{2\pi\sigma^2}} \int_0^{\infty} \frac{1}{\sqrt{1+\nu^2}} e^{-\frac{y^2}{2\sigma^2(1+\nu^2)}} p(\nu) d\nu \\
&= \frac{2}{\pi\sqrt{2\pi\sigma^2}} \int_0^{\infty} e^{-\frac{y^2}{2\sigma^2(1+\nu^2)}} \frac{1}{(1+\nu^2)^{3/2}} d\nu.
\end{aligned}
$$

Using the change of variable $u = \frac{1}{1+\nu^2}$, we obtain

$$
\begin{aligned}
\pi(y) &= \frac{1}{\pi\sqrt{2\pi\sigma^2}} \int_0^1 e^{-\frac{uy^2}{2\sigma^2}} (1-u)^{-1/2} du \\
&= \frac{2}{\pi\sqrt{2\pi\sigma^2}} \, {}_1F_1\left(1, \frac{3}{2}, -\frac{y^2}{2\sigma^2}\right),
\end{aligned}
$$

where ${}_1F_1$ is (Kummer's) confluent hypergeometric function of the first kind, with integral representation

$$
{}_1F_1(a, b, z) = \frac{\Gamma(b)}{\Gamma(a)\Gamma(b-a)} \int_0^1 e^{zt} t^{a-1} (1-t)^{b-a-1} dt.
$$

Alternatively, the marginal can be expressed in function of the imaginary error function (erfi) or Dawson function (aka Dawson integral) as

$$
\begin{aligned}
\pi(y) &= \frac{1}{\pi\sqrt{2\sigma^2}} e^{-y^2/(2\sigma^2)} \frac{\mathrm{erfi}(|y|/\sqrt{2\sigma^2})}{|y|/(\sqrt{2\sigma^2})} \\
&= \frac{2}{\pi^{3/2}} \frac{1}{|y|} D\left(\frac{|y|}{\sqrt{2\sigma^2}}\right)
\end{aligned}
$$

where Dawson's function is defined as

$$D(z) = e^{-z^2} \int_0^z e^{t^2} dt.$$

The marginal likelihood exhibits power-law tails

$$\pi(y) \sim C \frac{1}{|y|^2} \quad \text{as } |y| \to \infty$$

for some constant $C > 0$. Let $\ell(y) = \log \pi(y)$ denote the log-marginal likelihood. Kummer's function has the derivative

$$\frac{d}{dz} {}_1F_1(a, b, z) = \frac{a}{b} {}_1F_1(a+1, b+1, z).$$

It follows that

$$\ell'(y) = \frac{\pi'(y)}{\pi(y)}$$

$$= -\frac{2}{3}\frac{y}{\sigma^2}\frac{{}_1F_1\left(2,\frac{5}{2},-\frac{y^2}{2\sigma^2}\right)}{{}_1F_1\left(1,\frac{3}{2},-\frac{y^2}{2\sigma^2}\right)}.$$

Applying Tweedie's formula (Efron, 2011), we obtain the posterior mean

$$\mathbb{E}[\beta \mid y] = y + \sigma^2\ell'(y;\sigma) = (1 - \kappa(y))y, \tag{S27}$$

where the shrinkage function $\kappa(y) \in (0,1)$ is given by

$$\kappa(y) = \frac{2}{3}\frac{{}_1F_1\left(2,\frac{5}{2},-\frac{y^2}{2\sigma^2}\right)}{{}_1F_1\left(1,\frac{3}{2},-\frac{y^2}{2\sigma^2}\right)}.$$

Using the asymptotic expansion (Slater, 1960, Chapter 4, Eq. (4.I.3))

$$_1F_1(a,b,-z) \sim z^{-a}\frac{\Gamma(b)}{\Gamma(b-a)}$$

as $z \to \infty$, we find

$$\kappa(y) \sim \frac{2\sigma^2}{y^2}$$

$$|\mathbb{E}[\beta \mid y] - y| = \sigma^2|\ell'(y)| \sim \frac{2\sigma^2}{|y|}$$

as $|y| \to \infty$.

The horseshoe prior $\pi_{\mathrm{HS}}$ has two key properties: an infinite spike at zero, inducing strong shrinkage near $y = 0$, and Cauchy-like tails, ensuring that strong signals remain largely unshrunk ($\kappa(y) \to 0$ and $|\mathbb{E}[\beta \mid y] - y| \to 0$ as $|y| \to \infty$). This is illustrated in Figure S4.

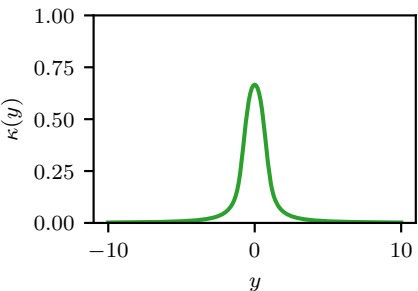

*Figure S4.* Shrinkage function $\kappa(y)$ for the horseshoe prior when $\sigma^2 = 0.1$.

*Remark* S1.1 (Parameterisation). In this section and in the main text, we focused on the specific parameterisation $\eta = 1$. For a general $\eta$, similar expressions can be derived for the marginal likelihood and posterior mean, replacing Kummer's $_1F_1$ function with the more general degenerate hypergeometric function of two variables, $\Phi_1$ (see (Carvalho et al., 2010, Equations (4) in the main text and (A1) in the appendix)). While Kummer's $_1F_1$ function is implemented in many standard scientific libraries, such as SciPy, $\Phi_1$ is not. Consequently, computing the marginal likelihood when $\eta \neq 1$ requires numerical integration. Since the evaluation of the marginal likelihood is crucial to our approach, it is therefore reasonable to set $\eta = 1$ here.

## S1.2. FAB Framework

In this section, we provide additional background on the FAB framework (Pratt, 1961; 1963; Yu & Hoff, 2018).

Let $Y \mid \beta \sim \mathcal{N}(\beta, \sigma^2)$ with some prior $\pi_0(\beta)$. Denote by $\pi(y) = \int_{\mathbb{R}} p(y \mid \beta)\pi_0(\beta)d\beta$ the corresponding marginal likelihood. For $\alpha \in (0, 1)$, let $\mathcal{C}_\alpha$ be the confidence procedure that solves the constrained optimisation problem

$$\mathcal{C}_\alpha = \arg \min_{\widetilde{\mathcal{C}}_\alpha} \mathbb{E}[\mathrm{vol}(\widetilde{\mathcal{C}}_\alpha(Y))]$$

under the constraints $\Pr(\beta \in \mathcal{C}_\alpha(Y) \mid \beta = \beta') = 1 - \alpha$ for all fixed $\beta'$,

where $\mathrm{vol}(\mathcal{C}_\alpha(y)) = \int_{\beta' \in \mathcal{C}_\alpha(y)} d\beta'$ is the volume of $\mathcal{C}_\alpha(y)$ and

$$\mathbb{E}[\mathrm{vol}(\mathcal{C}_\alpha(Y))] = \int_{\mathbb{R}} \mathrm{vol}(\mathcal{C}_\alpha(y))\pi(y)dy \tag{S28}$$

is the expected volume under the marginal distribution $\pi(y)$. By the Ghosh-Pratt identity (Ghosh, 1961; Pratt, 1961),

$$\begin{aligned}
\mathbb{E}[\mathrm{vol}(\mathcal{C}_\alpha(Y))] &= \int_{\mathbb{R}} \mathrm{vol}(\mathcal{C}_\alpha(y))\pi(y)dy \\
&= \int_{\mathbb{R}} \int_{\mathbb{R}} 1_{\beta' \in \mathcal{C}_\alpha(y)} d\beta' \pi(y)dy \\
&= \int_{\mathbb{R}} \Pr(\beta' \in \mathcal{C}_\alpha(Y))d\beta'.
\end{aligned}$$

That is, minimising $\mathbb{E}[\mathrm{vol}(\mathcal{C}_\alpha(Y))]$ is equivalent to minimising $\Pr(\beta' \in \mathcal{C}_\alpha(Y))$ for each $\beta' \in \mathbb{R}$. Define the acceptance region

$$A_\alpha(\beta') = \{y \mid \beta' \in \mathcal{C}_\alpha(y)\}.$$

The constrained optimisation problem above then reduces to solving, for each $\beta'$,

$$A_\alpha(\beta') = \arg \max_{\widetilde{A}_\alpha} \Pr(Y \notin \widetilde{A}_\alpha(\beta'))$$

such that $\Pr(Y \notin A_\alpha(\beta) \mid \beta = \beta') = \alpha$.

The term $\Pr(Y \notin A_\alpha(\beta'))$ may be interpreted as the power of a size-$\alpha$ test

$$H_0 : \beta = \beta' \text{ vs } H_1 : \beta \sim \pi_0,$$

where $Y \mid \beta \sim \mathcal{N}(\beta, \sigma^2)$. By the Neyman-Pearson lemma, the most powerful test is of the form

$$A_\alpha(\beta') = \left\{ y \mid \frac{\pi(y)}{p(y \mid \beta')} \leq k_\alpha(\beta') \right\}$$

where $k_\alpha(\beta')$ is such that $\Pr(Y \in A_\alpha(\beta) \mid \beta = \beta') = 1 - \alpha$. The acceptance region is an interval $[\underline{A}_\alpha(\beta'), \overline{A}_\alpha(\beta')]$ (Cortinovis & Caron, 2024, Theorem 3.3). Defining

$$w_\alpha(\beta') = \frac{1}{\alpha}\Phi\left(\frac{\underline{A}_\alpha(\beta') - \beta'}{\sigma}\right),$$

the confidence region is given by

$$\mathcal{C}_\alpha(y) = \{\beta' \mid \underline{A}_\alpha(\beta') = \beta' - \sigma z_{1-\alpha w_\alpha(\beta')} \leq y \leq \beta' + \sigma z_{1-\alpha(1-w_\alpha(\beta'))} = \overline{A}_\alpha(\beta')\}. \tag{S29}$$

The function $w_\alpha(\beta') \in [0, 1]$ is called the spending function or tail function (Puza & O'Neill, 2006; Yu & Hoff, 2018), and represents the proportion of the $\alpha$ rejection budget allocated to the left tail of the acceptance interval $[\underline{A}_\alpha(\beta'), \overline{A}_\alpha(\beta')]$.

The spending function $w_\alpha$ satisfies several key properties, which will be useful for our asymptotic analysis. Most of these originate from Cortinovis & Caron (2024). Under mild assumptions on the prior, satisfied for the models considered in this paper, $w_\alpha(\beta)$ is continuous in $\beta$. If the prior $\pi_0$ is symmetric around zero, we have

$$w_\alpha(-\beta') = 1 - w_\alpha(\beta'). \tag{S30}$$

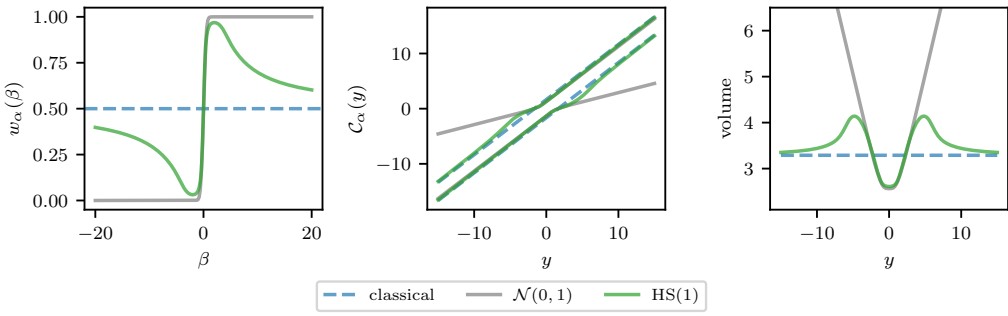

*Figure S5.* Comparison of the FAB procedures under a Gaussian ($\tau^2 = 1$) and a horseshoe ($\eta = 1$) priors when $\sigma^2 = 1$ and $\alpha = 0.1$.

Additionally, if the prior $\pi_0(\beta) := \pi_0(\beta; \sigma)$ admits $\sigma$ as a scale parameter, writing $w_\alpha(\beta; \sigma)$ for the corresponding tail function, we have

$$w_\alpha(\beta; \sigma) = w_\alpha\left(\frac{\beta}{\sigma}; 1\right). \tag{S31}$$

We now describe other properties of the spending function in the case of a Gaussian prior and of a prior with power-law tails, such as the horseshoe.

**Proposition S1.2** (FAB with a Gaussian prior (Pratt, 1963; Yu & Hoff, 2018))**.** *If the prior $\pi_0(\beta) = \mathcal{N}(\beta; 0, \tau^2\sigma^2)$ is Gaussian, the spending function is given by $w_\alpha(\beta) = g_\alpha^{-1}\left(\frac{2\beta}{\sigma\tau^2}\right)$, where $g_\alpha : (0, 1) \to \mathbb{R}$ is the one-to-one function*

$$g_\alpha(\omega) = \Phi^{-1}(\alpha\omega) - \Phi^{-1}(\alpha(1 - \omega)). \tag{S32}$$

*$w_\alpha$ is strictly increasing and*

$$\lim_{\beta \to \infty} w_\alpha(\beta) = 1.$$

**Proposition S1.3** (FAB with a prior with power-law tails (Cortinovis & Caron, 2024, Lemma S1.1))**.** *Let $\pi_0(\beta; \sigma)$ be a symmetric prior on $\beta$ such that the marginal density $\pi(y)$ has power-law tails, i.e.*

$$\pi(y) \sim C_\sigma |y|^{-\delta} \text{ as } |y| \to \infty$$

*for some constant $C_\sigma$ and some exponent $\delta > 1$. Then, $w_\alpha(\beta; \sigma)$ is bounded away from $0$ and $1$, and*

$$\lim_{\beta \to \infty} w_\alpha(\beta) = \lim_{\beta \to -\infty} w_\alpha(\beta) = \frac{1}{2}.$$

The difference between the spending functions of the two priors greatly affects the resulting FAB confidence regions. In particular, while both priors lead to confidence regions that are shorter than the classical one when the observed $y$ is close to zero, their behaviour differs as the disagreement between the prior and the data increases. In particular, the FAB confidence regions under the Gaussian prior become unbounded as $|y|$ grows, while the horseshoe prior leads to confidence regions that eventually revert to the classical confidence interval. This is illustrated in Figure S5.

## S2. Derivations

### S2.1. FAB-PPI for Mean Estimation

Here, we outline the steps to derive the FAB-PPI mean estimator presented in Equation (24), as well as the corresponding FAB-PPI confidence region.

The convex loss function that corresponds to estimating $\theta^\star = \mathbb{E}[Y]$ is the squared loss $\mathcal{L}_\theta(x, y) = \frac{1}{2}(\theta - y)^2$. In this case, the subgradient of $\mathcal{L}_\theta$ with respect to $\theta$ is given by $\mathcal{L}'_\theta(x, y) = \theta - y$. As a result of this, the measure of fit $m_\theta$ and the

rectifier $\Delta_\theta$ take the form

$$m_\theta = \mathbb{E}[\mathcal{L}'_\theta(X, f(X))] = \theta - \mathbb{E}[f(X)],$$
$$\Delta_\theta = \mathbb{E}[\mathcal{L}'_\theta(X, Y) - \mathcal{L}'_\theta(X, f(X))] = \mathbb{E}[f(X) - Y].$$

In particular, under the squared loss, the rectifier $\Delta_\theta$ does not depend on $\theta$, and we indicate this by dropping the subscript $\theta$ and writing $\Delta := \Delta_\theta$.

In order to apply FAB-PPI to this setting, we follow the steps outlined in Section 4. In particular, we use the sample mean of the unlabelled data (9) as the estimator $\widehat{m}_\theta$ of $m_\theta$,

$$\widehat{m}_\theta = \frac{1}{N} \sum_{i=1}^{N} \mathcal{L}'_\theta(\widetilde{X}_i, f(\widetilde{X}_i)) = \theta - \frac{1}{N} \sum_{i=1}^{N} f(\widetilde{X}_i),$$

and either the sample mean (10) or the control variate estimator (12) as the estimator $\widehat{\Delta}$ of $\Delta$, as in PPI and PPI++, respectively. To avoid repetitions, in this section we write $\widehat{\Delta}$ as the following general control variate estimator with tuning parameter $\lambda \in \mathbb{R}$,

$$\widehat{\Delta} = \frac{1}{n} \sum_{i=1}^{n} (\mathcal{L}'_\theta(X_i, Y_i) - \mathcal{L}'_\theta(X_i, f(X_i))) - (\lambda - 1) \left( \frac{1}{n} \left[ \sum_{i=1}^{n} \mathcal{L}'_\theta(X_i, f(X_i)) \right] - \widehat{m}_\theta \right)$$

$$= \frac{1}{n} \sum_{i=1}^{n} (\mathcal{L}'_\theta(X_i, Y_i) - \lambda \mathcal{L}'_\theta(X_i, f(X_i))) + (\lambda - 1) \widehat{m}_\theta$$

$$= -\overline{Y} + \lambda \left( \frac{1}{n} \sum_{i=1}^{n} f(X_i) - \frac{1}{N} \sum_{j=1}^{N} f(\widetilde{X}_j) \right) + \frac{1}{N} \sum_{j=1}^{N} f(\widetilde{X}_j).$$

The sample mean estimator (10) and the control variate estimator (12) under the squared loss are recovered by setting $\lambda$ to 1 and $\widehat{\lambda}$ as in Equation (14), respectively. From this, the standard PPI mean estimators (11) and (13) are obtained by solving the equation $\widehat{m}_\theta + \widehat{\Delta} = 0$ for $\theta$.

Instead, we first define the Bayes-assisted estimator (20) under the chosen prior $\pi_0(\Delta; \tau_n)$,

$$\widehat{\Delta}^{\text{FABPP}} = \widehat{\Delta} + \widehat{\sigma}^2 \ell' \left( \widehat{\Delta}; \widehat{\sigma}, \tau_n \right),$$

where $\widehat{\sigma}^2$ is an estimator of $\text{var}(\widehat{\Delta})$ and $\ell'_\theta(z; \sigma, \tau)$ is the derivative of the log-marginal likelihood of a Gaussian likelihood model with mean $\Delta$ and variance $\sigma^2$ under the prior $\pi_0(\Delta, \tau)$. Then, the FAB-PPI mean estimator $\widehat{\theta}^{\text{FABPP}}$ under $\pi_0$ is given by the solution to the equation

$$\widehat{m}_\theta + \widehat{\Delta}^{\text{FABPP}} = 0$$

in $\theta$, that takes the form

$$\widehat{\theta}^{\text{FABPP}} = \overline{Y} - \lambda \left( \frac{1}{n} \sum_{i=1}^{n} f(X_i) - \frac{1}{N} \sum_{j=1}^{N} f(\widetilde{X}_j) \right) - \widehat{\sigma}^2 \ell' \left( \widehat{\Delta}; \widehat{\sigma}, \tau_n \right),$$

which matches the expression in Equation (24). Furthermore, by recognising that the first two terms in the above expression match (13), we can alternatively write the FAB-PPI mean estimator as

$$\widehat{\theta}^{\text{FABPP}} = \widehat{\theta} - \widehat{\sigma}^2 \ell' \left( \widehat{\Delta}; \widehat{\sigma}, \tau_n \right),$$

where $\widehat{\theta}$ is the corresponding standard PPI mean estimator.

Given $\alpha \in (0, 1)$, we construct the FAB-PPI confidence region $\mathcal{C}_\alpha^{\text{FABPP}}$ for the mean as described in Section 4.2. In particular, let $\mathcal{T}_{\alpha-\delta}(\widehat{m}_\theta)$ denote a standard $1 - (\alpha - \delta)$ confidence interval for $m_\theta$,

$$\mathcal{T}_{\alpha-\delta}(\widehat{m}_\theta) = [\widehat{m}_\theta \pm \widehat{\sigma}^f z_{1-(\alpha-\delta)/2}] = \left[ \theta - \widehat{\theta}^f \pm \widehat{\sigma}^f z_{1-(\alpha-\delta)/2} \right],$$

where $\widehat{\theta}^f := \frac{1}{N}\sum_{i=1}^N f(\widetilde{X}_i)$ for conciseness, and $(\widehat{\sigma}^f)^2$ is an estimator of $\mathrm{var}(\widehat{m}_\theta)$. Then, we apply the FAB framework under the prior $\pi_0(\Delta; \tau_n)$ to obtain a $1 - \delta$ confidence region for $\Delta$,

$$\mathcal{R}_\delta^{\mathrm{FABPP}}(\widehat{\Delta}) = \mathrm{FAB\text{-}CR}(\widehat{\Delta}; \pi_0(\cdot\,; \tau_n), \widehat{\sigma}, \delta),$$

where, again, $\widehat{\sigma}^2$ is an estimator of $\mathrm{var}(\widehat{\Delta})$. Finally, to avoid making assumptions on the specific form of $\mathcal{R}_\delta^{\mathrm{FABPP}}(\widehat{\Delta})$, we use $[\inf \mathcal{R}_\delta^{\mathrm{FABPP}}), \sup(\mathcal{R}_\delta^{\mathrm{FABPP}})] \supseteq \mathcal{R}_\delta^{\mathrm{FABPP}}(\widehat{\Delta})$ in the definition of $\mathcal{C}_\alpha^{\mathrm{FABPP}}$ (21) to obtain the FAB-PPI interval

$$
\begin{aligned}
\mathcal{C}_\alpha^{\mathrm{FABPP}} &= \left\{ \theta \mid 0 \in \left[ \theta - \widehat{\theta}^f \pm \widehat{\sigma}^f z_{1-(\alpha-\delta)/2} \right] + [\inf(\mathcal{R}_\delta^{\mathrm{FABPP}}), \sup(\mathcal{R}_\delta^{\mathrm{FABPP}})] \right\} \\
&= \left\{ \theta \mid 0 \in \left[ \theta - \widehat{\theta}^f - \widehat{\sigma}^f z_{1-(\alpha-\delta)/2} + \inf(\mathcal{R}_\delta^{\mathrm{FABPP}}), \theta - \widehat{\theta}^f + \widehat{\sigma}^f z_{1-(\alpha-\delta)/2} + \sup(\mathcal{R}_\delta^{\mathrm{FABPP}}) \right] \right\} \\
&= \left[ \widehat{\theta}^f - \widehat{\sigma}^f z_{1-(\alpha-\delta)/2} - \sup(\mathcal{R}_\delta^{\mathrm{FABPP}}), \widehat{\theta}^f + \widehat{\sigma}^f z_{1-(\alpha-\delta)/2} - \inf(\mathcal{R}_\delta^{\mathrm{FABPP}}) \right].
\end{aligned}
$$

Algorithm 2 summarises the FAB-PPI approach under the squared loss, where $\widehat{\xi}$ is defined for notational convenience and the corresponding sample variances are used as $(\widehat{\sigma}^f)^2$ and $\widehat{\sigma}^2$.

---

**Algorithm 2** FAB-PPI for mean estimation

---

**Input:** labelled $\{(X_i, Y_i)\}_{i=1}^n$, unlabelled $\{\widetilde{X}_j\}_{j=1}^N$, predictor $f$, prior $\pi_0(\cdot\,; \tau_n)$, error levels $\alpha, \delta$

Set $\widehat{\lambda} = 1$ (FAB-PPI) or estimate $\widehat{\lambda}$ from data (FAB-PPI++) using Equation (14)
$\widehat{\theta}^f \leftarrow \frac{1}{N} \sum_{j=1}^N f(\widetilde{X}_j)$
$\widehat{\xi} \leftarrow \frac{1}{n} \sum_{i=1}^n (\widehat{\lambda} f(X_i) - Y_i)$
$\widehat{\Delta} \leftarrow \widehat{\xi} - (\widehat{\lambda} - 1)\widehat{\theta}^f$
$(\widehat{\sigma}^f)^2 \leftarrow \frac{1}{N(N-1)} \sum_{j=1}^N (f(\widetilde{X}_j) - \widehat{\theta}^f)^2$
$\widehat{\sigma}_\xi^2 = \frac{1}{n-1} \sum_{i=1}^n (\widehat{\lambda} f(X_i) - Y_i - \widehat{\xi})^2$
$\widehat{\sigma}^2 \leftarrow \frac{1}{n} \widehat{\sigma}_\xi^2 + (\widehat{\lambda} - 1)^2 (\widehat{\sigma}^f)^2$
$\mathcal{R}_\delta^{\mathrm{FABPP}} \leftarrow \mathrm{FAB\text{-}CR}(\widehat{\Delta}; \pi_0(\cdot\,; \tau_n), \widehat{\sigma}, \delta)$
**Outputs:** estimator $\widehat{\theta}^{\mathrm{FABPP}} = \widehat{\theta}^f - \widehat{\Delta} - \widehat{\sigma}^2 \ell'(\widehat{\Delta}; \widehat{\sigma}, \tau_n)$ and CR $\mathcal{C}_\alpha^{\mathrm{FABPP}} = [\widehat{\theta}^f - \widehat{\sigma}^f z_{1-(\alpha-\delta)/2} - \sup(\mathcal{R}_\delta^{\mathrm{FABPP}}), \widehat{\theta}^f + \widehat{\sigma}^f z_{1-(\alpha-\delta)/2} - \inf(\mathcal{R}_\delta^{\mathrm{FABPP}})]$

---

# S3. Proofs

Some of the results discussed in this section, such as Lemma S3.2 and Corollary S3.5, are concerned with the convergence of closed sets with respect to the Hausdorff distance, which we recall here for completeness. Given two closed subsets $C_1$ and $C_2$ of $\mathbb{R}$, their Hausdorff distance $d_\mathrm{H}$ is defined as

$$d_\mathrm{H}(C_1, C_2) = \max \left\{ \sup_{x \in C_1} \inf_{y \in C_2} |x - y|, \sup_{y \in C_2} \inf_{x \in C_1} |x - y| \right\}.$$

Then, for a collection of closed subsets $(C_1(y))_{y \in \mathbb{R}}$ and a closed subset $C_2$ of $\mathbb{R}$, $(C_1(y))_{y \in \mathbb{R}}$ converges in Hausdorff distance to $C_2$ if $\lim_{y \to \infty} d_\mathrm{H}(C_1(y), C_2) = 0$. In particular, if $C_2 = [a, b]$ is a closed interval for some $a < b$, $\lim_{y \to \infty} d_\mathrm{H}(C_1(y), C_2) = 0$ if and only if, for all $\epsilon \in (0, \frac{b-a}{2})$, there exists $y_0$ such that $[a+\epsilon, b-\epsilon] \subseteq C_1(y) \subseteq [a-\epsilon, b+\epsilon]$ for all $y > y_0$. In the sequel, we write $\lim_{y \to \infty} C_1(y) = C_2$ for $\lim_{y \to \infty} d_\mathrm{H}(C_1(y), C_2) = 0$.

## S3.1. Theorem 4.1 - Asymptotic Coverage of FAB-PPI under the Gaussian and Horseshoe Priors

Under the prior $\pi_0(\cdot; \widehat{\sigma}_\theta)$, the FAB confidence region for the rectifier $\Delta_\theta$ is

$$\mathcal{R}_\delta^{\mathrm{FABPP}}(\widehat{\Delta}_\theta; \widehat{\sigma}_\theta) = \{\Delta_\theta \mid \Delta_\theta - \widehat{\sigma}_\theta z_{1-\delta w_\delta(\Delta_\theta; \widehat{\sigma}_\theta)} \leq \widehat{\Delta}_\theta \leq \Delta_\theta + \widehat{\sigma}_\theta z_{1-\delta(1-w_\delta(\Delta_\theta; \widehat{\sigma}_\theta))}\}, \tag{S33}$$

where $w_\delta(\cdot; \widehat{\sigma}_\theta)$ is the FAB spending function. The proof of asymptotic coverage is organised as follows. First, we show that $\mathcal{R}_\delta^{\mathrm{FABPP}}(\widehat{\Delta}_\theta; \widehat{\sigma}_\theta)$ is asymptotically a $1 - \delta$ confidence interval. This is established via Lemma S3.1 for the Gaussian prior,

and via Lemma S3.2 for the horseshoe prior. This result is then combined with the asymptotic coverage of the standard sample mean estimator for $m_\theta$ to conclude the asymptotic coverage of the FAB-PPI estimator of $\theta^\star$.

We first prove the following lemma for the Gaussian prior, demonstrating that the rectifier has the correct asymptotic coverage.

**Lemma S3.1.** *Let $\widehat{\Delta}_\theta$ be a consistent estimator of $\Delta_\theta$ such that a CLT holds for $\widehat{\Delta}_\theta$, i.e.*

$$\frac{\widehat{\Delta}_\theta - \Delta_\theta}{\widehat{\sigma}_\theta} \to \mathcal{N}(0, 1)$$

*as $\min(n, N) \to \infty$, where $\frac{\widehat{\sigma}_\theta^2}{\mathrm{var}(\widehat{\Delta}_\theta)} \to 1$ almost surely. Let $\pi_0(\cdot; \widehat{\sigma}_\theta)$ be the Gaussian prior (22) for $\Delta_\theta$ and consider the corresponding $1 - \delta$ FAB confidence region*

$$\mathcal{R}_\delta^{FABPP}(\widehat{\Delta}_\theta; \widehat{\sigma}_\theta) = \textit{FAB-CR}\left(\widehat{\Delta}_\theta; \pi_0\left(\cdot\,; \widehat{\sigma}_\theta\right), \widehat{\sigma}_\theta, \delta\right).$$

*Then*

$$\liminf_{\min(n,N)\to\infty} \Pr(\Delta_\theta \in \mathcal{R}_\delta^{FABPP}(\widehat{\Delta}_\theta; \widehat{\sigma}_\theta) \mid \Delta_\theta) \geq 1 - \delta. \tag{S34}$$

*Proof.* Using Proposition S1.2, for any $x > 0$, $\sigma > 0$,

$$\begin{aligned}
\frac{2x}{\sigma} &= g_\delta(g_\delta^{-1}(2x/\sigma)) \\
&= g_\delta(w_\delta(x; \sigma)) \\
&= z_{1-\delta(1-w_\delta(x;\sigma))} - z_{1-\delta w_\delta(x;\sigma)}
\end{aligned}$$

and

$$\begin{aligned}
\sigma z_{1-\delta(1-w_\delta(x;\sigma))} &= \sigma z_{1-\delta(1-w_\delta(x/\sigma;1))} \\
&= 2x + \sigma z_{1-\delta w_\delta(x/\sigma;1)}.
\end{aligned}$$

The FAB confidence region (S33) can therefore be written as

$$\begin{aligned}
\mathcal{R}_\delta^{FABPP}(\widehat{\Delta}_\theta; \widehat{\sigma}_\theta) = &\{\Delta_\theta > 0 \mid \Delta_\theta - \widehat{\sigma}_\theta z_{1-\delta w_\delta(\Delta_\theta;\widehat{\sigma}_\theta)} \leq \widehat{\Delta}_\theta \leq 3\Delta_\theta + \widehat{\sigma}_\theta z_{1-\delta w_\delta(\Delta_\theta;\widehat{\sigma}_\theta)}\} \\
&\cup \{\Delta_\theta < 0 \mid 3\Delta_\theta - \widehat{\sigma}_\theta z_{1-\delta(1-w_\delta(\Delta_\theta;\widehat{\sigma}_\theta))} \leq \widehat{\Delta}_\theta \leq \Delta_\theta + \widehat{\sigma}_\theta z_{1-\delta(1-w_\delta(\Delta_\theta;\widehat{\sigma}_\theta))}\} \\
&\cup \{0 \mid |\widehat{\Delta}_\theta| \leq \widehat{\sigma}_\theta z_{1-\delta/2}\}.
\end{aligned}$$

Consider first $\Delta_\theta = 0$. By the CLT, $\Pr(0 \in \mathcal{R}_\delta^{FABPP}(\widehat{\Delta}_\theta; \widehat{\sigma}_\theta) \mid \Delta_\theta = 0) = \Pr(|\widehat{\Delta}_\theta| \leq \widehat{\sigma}_\theta z_{1-\delta/2} \mid \Delta_\theta = 0) \to 1 - \delta$ as $\min(n, N) \to \infty$. Additionally, for any $x > 0$,

$$\begin{aligned}
z_{1-\delta w_\delta(x;\widehat{\sigma}_\theta)} &\to z_{1-\delta} \\
z_{1-\delta(1-w_\delta(-x;\widehat{\sigma}_\theta))} &\to z_{1-\delta}
\end{aligned}$$

almost surely as $\min(n, N) \to \infty$.

It follows from Equation (S33) that, for any $\epsilon \in (0, z_{1-\delta})$, there exist $N_0$ such that for all $n, N$ with $\min(n, N) \geq N_0$, the FAB confidence region $\mathcal{R}_\delta^{FABPP}(\widehat{\Delta}_\theta; \widehat{\sigma}_\theta)$ contains the set

$$\begin{aligned}
\mathcal{S}_\delta(\widehat{\Delta}_\theta; \widehat{\sigma}_\theta) = &\left\{\Delta_\theta > 0 \mid \Delta_\theta - \widehat{\sigma}_\theta(z_{1-\delta} - \epsilon) \leq \widehat{\Delta}_\theta \leq 3\Delta_\theta + \widehat{\sigma}_\theta(z_{1-\delta} - \epsilon)\right\} \\
&\cup \left\{\Delta_\theta < 0 \mid 3\Delta_\theta - \widehat{\sigma}_\theta(z_{1-\delta} - \epsilon) \leq \widehat{\Delta}_\theta \leq \Delta_\theta - \widehat{\sigma}_\theta(z_{1-\delta} - \epsilon)\right\} \cup \{0 \mid |\widehat{\Delta}_\theta| \leq \widehat{\sigma}_\theta z_{1-\delta/2}\}.
\end{aligned}$$

For any fixed $\Delta_\theta > 0$,

$$\Pr(\Delta_\theta \in \mathcal{S}_\delta(\widehat{\Delta}_\theta; \widehat{\sigma}_\theta)) = \Pr\left(-(z_{1-\delta} - \epsilon) \leq \frac{\widehat{\Delta}_\theta - \Delta_\theta}{\widehat{\sigma}_\theta} \leq \frac{2\Delta_\theta}{\widehat{\sigma}_\theta} + z_{1-\delta} - \epsilon\right). \tag{S35}$$

Noting that $\frac{2\Delta_\theta}{\widehat{\sigma}_\theta} + z_{1-\delta} - \epsilon \to \infty$ a.s. as $\min(n, N) \to \infty$, we obtain that

$$\liminf_{\min(n,N)\to\infty} \Pr(\Delta_\theta \in \mathcal{R}_\delta^{\text{FABPP}}(\widehat{\Delta}_\theta; \widehat{\sigma}_\theta) \mid \Delta_\theta) \geq \liminf_{\min(n,N)\to\infty} \Pr(\Delta_\theta \in \mathcal{S}_\delta(\widehat{\Delta}_\theta; \widehat{\sigma}_\theta) \mid \Delta_\theta) \geq 1 - \delta. \quad \text{(S36)}$$

The proof proceeds similarly for $\Delta_\theta < 0$. □

**Lemma S3.2.** *Let $\widehat{\Delta}_\theta$ be a consistent estimator of $\Delta_\theta$ such that a CLT holds for $\widehat{\Delta}_\theta$, i.e.*

$$\frac{\widehat{\Delta}_\theta - \Delta_\theta}{\widehat{\sigma}_\theta} \to \mathcal{N}(0, 1)$$

*as $\min(n, N) \to \infty$, where $\widehat{\sigma}_\theta^2/\text{var}(\widehat{\Delta}_\theta) \to 1$ almost surely. Let the prior $\pi_0(\cdot; \widehat{\sigma}_\theta)$ on $\Delta_\theta$ be the horseshoe prior (23) with scale parameter $\widehat{\sigma}_\theta$ and consider the corresponding $1 - \delta$ FAB confidence region*

$$\mathcal{R}_\delta^{FABPP}(\widehat{\Delta}_\theta; \widehat{\sigma}_\theta) = \text{FAB-CR}\left(\widehat{\Delta}_\theta; \pi_0(\cdot; \widehat{\sigma}_\theta), \widehat{\sigma}_\theta, \delta\right),$$

*where $w_\delta(\Delta_\theta; \widehat{\sigma}_\theta)$ is the associated weight function. Then, for $\Delta_\theta \neq 0$, the confidence region $\mathcal{R}_\delta^{FABPP}(\widehat{\Delta}_\theta; \widehat{\sigma}_\theta)$ reverts to the classical $1 - \delta$ z-interval for $\Delta_\theta$, i.e., almost surely,*

$$\lim_{\min(n,N)\to\infty} \frac{\mathcal{R}_\delta^{FABPP}(\widehat{\Delta}_\theta; \widehat{\sigma}_\theta) - \widehat{\Delta}_\theta}{\widehat{\sigma}_\theta} = [-z_{1-\delta/2}, z_{1-\delta/2}],$$

*where the convergence is with respect to the Hausdorff distance on closed subsets of $\mathbb{R}$. Moreover, for any $\Delta_\theta \in \mathbb{R}$,*

$$\lim_{\min(n,N)\to\infty} \Pr(\Delta_\theta \in \mathcal{R}_\delta^{FABPP}(\widehat{\Delta}_\theta; \widehat{\sigma}_\theta) \mid \Delta_\theta) = 1 - \delta. \quad \text{(S37)}$$

*Proof.* Consider the case $\Delta_\theta \neq 0$. As described in Section S1.2, the spending function $w_\delta$ is continuous and satisfies, for any $z \in \mathbb{R}$ and $\sigma > 0$,

$$w_\delta(z; \sigma) = w_\delta\left(\frac{z}{\sigma}; 1\right). \quad \text{(S38)}$$

Moreover, by Proposition S1.3, $w_\delta(\cdot; 1)$ takes values in $(0, 1)$ and satisfies

$$\lim_{z\to\infty} w_\delta(z; 1) = \lim_{z\to-\infty} w_\delta(z; 1) = w_\delta(0; 1) = \frac{1}{2}.$$

Define $A(p) = -\Phi^{-1}(1 - \delta(1 - p))$ and $B(p) = \Phi^{-1}(1 - \delta p)$, where $\Phi(\cdot)$ is the CDF of the standard normal distribution. Then, from Equation (S33), we have that

$$\begin{aligned}
\frac{\mathcal{R}_\delta^{\text{FABPP}}(\widehat{\Delta}_\theta; \widehat{\sigma}_\theta) - \widehat{\Delta}_\theta}{\widehat{\sigma}_\theta} &= \left\{\psi \in \mathbb{R} \mid A(w_\delta(\widehat{\sigma}_\theta\psi + \widehat{\Delta}_\theta; \widehat{\sigma}_\theta)) \leq \psi \leq B(w_\delta(\widehat{\sigma}_\theta\psi + \widehat{\Delta}_\theta; \widehat{\sigma}_\theta))\right\} \\
&= \left\{\psi \in \mathbb{R} \mid A(w_\delta(\psi + \widehat{\Delta}_\theta/\widehat{\sigma}_\theta; 1)) \leq \psi \leq B(w_\delta(\psi + \widehat{\Delta}_\theta/\widehat{\sigma}_\theta; 1))\right\} \\
&=: \mathcal{C}_{n,N},
\end{aligned}$$

where the second equality follows from Equation (S38).

Assume that $\Delta_\theta > 0$, which ensures $\widehat{\Delta}_\theta/\widehat{\sigma}_\theta \to \infty$ almost surely as $\min(n, N) \to \infty$. The case $\Delta_\theta < 0$ follows similarly.

First, we show that there exists an $M$ independent of $n, N$ such that, for all $n, N \geq 1$, if $\psi \in \mathcal{C}_{n,N}$, then $\psi > M$. By the boundedness of $w_\delta(\cdot; 1)$, there exists $\kappa \in (0, \frac{1}{2})$ such that $w_\delta(x; 1) \in [\kappa, 1 - \kappa]$ for all $x \in \mathbb{R}$. Since $A(\cdot)$ is decreasing,

$$A(w_\delta(\psi + \widehat{\Delta}_\theta/\widehat{\sigma}_\theta; 1)) \geq A(1 - \kappa) := c > -\infty$$

for all $\psi \in \mathbb{R}$, $n, N \geq 1$. Pick $M < c$. For all $\psi \leq M$ we have, for all $n, N \geq 1$, $\psi \leq M < c \leq A(w_\delta(\psi + \widehat{\Delta}_\theta/\widehat{\sigma}_\theta; 1))$. Hence, $\psi \notin \mathcal{C}_{n,N}$. As a result, $\mathcal{C}_{n,N} \cap [M, \infty) = \mathcal{C}_{n,N}$.

Second, we show that, almost surely, $w_\delta(\psi + \widehat{\Delta}_\theta/\widehat{\sigma}_\theta; 1)$ converges to $1/2$ uniformly on $[M, \infty)$ as $\min(n, N) \to \infty$. Given that $w_\delta(\cdot; 1)$ is bounded and converges pointwise to $1/2$, we have that

$$g(t) := \sup_{y \geq t} \left| w_\delta(y; 1) - \frac{1}{2} \right| \to 0$$

as $t \to \infty$, where $g(\cdot)$ is continuous and nonincreasing. As a result of this, and since $\widehat{\Delta}_\theta/\widehat{\sigma}_\theta \to \infty$ almost surely as $\min(n, N) \to \infty$, we have that

$$\sup_{\psi \geq M} \left| w_\delta(\psi + \widehat{\Delta}_\theta/\widehat{\sigma}_\theta; 1) - \frac{1}{2} \right| = g\left(\widehat{\Delta}_\theta/\widehat{\sigma}_\theta + M\right) \to 0 \tag{S39}$$

almost surely as $\min(n, N) \to \infty$.

Lastly, we combine the previous two steps to show the almost sure Hausdorff convergence of $\mathcal{C}_{n,N}$. The functions $A(\cdot)$ and $B(\cdot)$ are continuous with $A(1/2) = -z_{1-\delta/2}$ and $B(1/2) = z_{1-\delta/2}$. Hence, for every $\epsilon \in (0, z_{1-\delta/2})$, there exists $\eta > 0$ such that

$$\left| p - \frac{1}{2} \right| \leq \eta \Rightarrow \left| A(p) + z_{1-\delta/2} \right| \leq \epsilon \text{ and } \left| B(p) - z_{1-\delta/2} \right| \leq \epsilon$$

By Equation (S39), there exists $N_1$ such that, for all $\min(n, N) \geq N_1$,

$$\sup_{\psi \geq M} \left| w_\delta(\psi + \widehat{\Delta}_\theta/\widehat{\sigma}_\theta; 1) - \frac{1}{2} \right| \leq \eta.$$

Then, uniformly for $\psi \geq M$,

$$A(w_\delta(\psi + \widehat{\Delta}_\theta/\widehat{\sigma}_\theta; 1)) \in [-z_{1-\delta/2} - \epsilon, -z_{1-\delta/2} + \epsilon]$$
$$B(w_\delta(\psi + \widehat{\Delta}_\theta/\widehat{\sigma}_\theta; 1)) \in [z_{1-\delta/2} - \epsilon, z_{1-\delta/2} + \epsilon]$$

and, for $\min(n, N) \geq N_1$,

$$\left[ -z_{1-\delta/2} + \epsilon, z_{1-\delta/2} - \epsilon \right] \subseteq \mathcal{C}_{n,N} \cap [M, \infty) \subseteq [-z_{1-\delta/2} - \epsilon, z_{1-\delta/2} + \epsilon],$$

which, combined with the first step, gives the desired result. Moreover, Equation (S37) then follows directly, which completes the proof.

On the other hand, in the case $\Delta_\theta = 0$, asymptotic coverage follows like in the proof of Lemma S3.1. In particular, from Equation (S33), we have that

$$\Pr(0 \in \mathcal{R}_\delta^{\text{FABPP}}(\widehat{\Delta}_\theta; \widehat{\sigma}_\theta) | \Delta_\theta = 0) = \Pr(|\widehat{\Delta}_\theta| \leq \widehat{\sigma}_\theta z_{1-\delta/2})$$

thanks to the fact $w_\delta(0; \sigma) = 1/2$, and the result follows from the CLT as $\min(n, N) \to \infty$. While this is not necessary to show asymptotic coverage, it is interesting to note that $\mathcal{C}_{n,N}$ does not converge to a deterministic limit almost surely when $\Delta_\theta = 0$. Instead, by showing continuity of the mapping $\widehat{\Delta}_\theta/\widehat{\sigma}_\theta \mapsto \mathcal{C}_{n,N}(\widehat{\Delta}_\theta/\widehat{\sigma}_\theta)$ and again exploiting the CLT, one may prove that $\mathcal{C}_{n,N}$ converges in distribution to the random FAB confidence region $\mathcal{C}(Y) - Y$, where $\mathcal{C}(y)$ is defined in Equation (S29), for a unit scale and a horseshoe prior, and $Y \sim \mathcal{N}(0, 1)$. □

With the above two lemmas, we can now prove Theorem 4.1, which we restate here in extended form.

**Theorem S3.3.** *Consider a convex estimation problem whose solution can be expressed as in Equation (2). For all $\theta \in \mathbb{R}$, define $\widehat{\Delta}_\theta$ and $\widehat{m}_\theta$ as in Section 4 and let*

$$\mathcal{R}_\delta^{\text{FABPP}}(\widehat{\Delta}_\theta; \widehat{\sigma}_\theta) = \textit{FAB-CR}\left(\widehat{\Delta}_\theta; \pi_0\left(\cdot; \widehat{\sigma}_\theta\right), \widehat{\sigma}_\theta, \delta\right),$$
$$\mathcal{T}_{\alpha-\delta}(\widehat{m}_\theta; \widehat{\sigma}_\theta^f) = \left[\widehat{m}_\theta \pm \widehat{\sigma}_\theta^f z_{1-(\alpha-\delta)/2}\right],$$

*where* $\frac{\widehat{\sigma}_\theta^2}{\operatorname{var}(\widehat{\Delta}_\theta)} \to 1$ *and* $\frac{(\widehat{\sigma}_\theta^f)^2}{\operatorname{var}(\widehat{m}_\theta)} \to 1$ *almost surely as* $\min(n, N) \to \infty$. *Then, the FAB-PPI confidence region* $\mathcal{C}_\alpha^{FABPP}$, *defined as*

$$\mathcal{C}_\alpha^{FABPP} = \left\{ \theta \mid 0 \in \mathcal{R}_\delta^{FABPP}(\widehat{\Delta}_\theta; \widehat{\sigma}_\theta) + \mathcal{T}_{\alpha-\delta}(\widehat{m}_\theta; \widehat{\sigma}_\theta^f) \right\},$$

*has correct asymptotic coverage, i.e. it satisfies*

$$\liminf_{\min(n,N) \to \infty} \Pr(\theta^\star \in \mathcal{C}_\alpha^{FABPP}) = \liminf_{\min(n,N) \to \infty} \Pr(0 \in \mathcal{R}_\delta^{FABPP}(\widehat{\Delta}_{\theta^\star}; \widehat{\sigma}_{\theta^\star}) + \mathcal{T}_{\alpha-\delta}(\widehat{m}_{\theta^\star}; \widehat{\sigma}_{\theta^\star}^f)) \geq 1 - \alpha.$$

*Proof.* By Lemma S3.1 (Gaussian prior) and Lemma S3.2 (horseshoe prior), $\mathcal{R}_\delta^{\text{FABPP}}(\widehat{\Delta}_{\theta^\star}; \widehat{\sigma}_{\theta^\star})$ is an asymptotically valid $1 - \delta$ confidence region for $\Delta_{\theta^\star}$, that is

$$\liminf_{\min(n,N) \to \infty} \Pr(\Delta_{\theta^\star} \in \mathcal{R}_\delta^{\text{FABPP}}(\widehat{\Delta}_{\theta^\star}; \widehat{\sigma}_{\theta^\star})) \geq 1 - \delta.$$

Similarly, the CLT for $\widehat{m}_\theta$ implies that

$$\liminf_{\min(n,N) \to \infty} \Pr(m_{\theta^\star} \in \mathcal{T}_{\alpha-\delta}(\widehat{m}_{\theta^\star}; \widehat{\sigma}_{\theta^\star}^f)) \geq 1 - (\alpha - \delta).$$

Consider the event

$$E = \{\Delta_{\theta^\star} \in \mathcal{R}_\delta^{\text{FABPP}}(\widehat{\Delta}_{\theta^\star}; \widehat{\sigma}_{\theta^\star})\} \cap \{m_{\theta^\star} \in \mathcal{T}_{\alpha-\delta}(\widehat{m}_{\theta^\star}; \widehat{\sigma}_{\theta^\star}^f)\}.$$

By Boole's inequality,

$$
\begin{aligned}
\liminf_{n,N \to \infty} \Pr(E) &\geq 1 - \limsup_{\min(n,N) \to \infty} \Pr(\{\Delta_{\theta^\star} \notin \mathcal{R}_\delta^{\text{FABPP}}(\widehat{\Delta}_{\theta^\star}; \widehat{\sigma}_{\theta^\star})\} \cup \{m_{\theta^\star} \notin \mathcal{T}_{\alpha-\delta}(\widehat{m}_{\theta^\star}; \widehat{\sigma}_{\theta^\star}^f)\}) \\
&\geq 1 - \limsup_{\min(n,N)} \Pr(\{\Delta_{\theta^\star} \notin \mathcal{R}_\delta^{\text{FABPP}}(\widehat{\Delta}_{\theta^\star}; \widehat{\sigma}_{\theta^\star})\}) - \limsup_{N \to \infty} \Pr(\{m_{\theta^\star} \notin \mathcal{T}_{\alpha-\delta}(\widehat{m}_{\theta^\star}; \widehat{\sigma}_{\theta^\star}^f)\}) \\
&\geq 1 - \delta - (\alpha - \delta) \\
&= 1 - \alpha.
\end{aligned}
$$

Furthermore, on the event $E$, we have that

$$0 = \Delta_{\theta^\star} + m_{\theta^\star} \in \mathcal{R}_\delta^{\text{FABPP}}(\widehat{\Delta}_{\theta^\star}; \widehat{\sigma}_{\theta^\star}) + \mathcal{T}_{\alpha-\delta}(\widehat{m}_{\theta^\star}; \widehat{\sigma}_{\theta^\star}^f),$$

where the first equality follows from Equation (2). As a result of this,

$$\liminf_{\min(n,N) \to \infty} \Pr(0 \in \mathcal{R}_\delta^{\text{FABPP}}(\widehat{\Delta}_{\theta^\star}; \widehat{\sigma}_{\theta^\star}) + \mathcal{T}_{\alpha-\delta}(\widehat{m}_{\theta^\star}; \widehat{\sigma}_{\theta^\star}^f)) \geq 1 - \alpha,$$

as desired. □

*Remark* S3.4. With both the Gaussian and horseshoe priors, we obtain asymptotic coverage. However, the asymptotic confidence regions differ significantly. In the Gaussian case, the volume of the confidence region does not vanish asymptotically. Instead, the confidence region converges to $(\frac{\Delta_\theta}{3}, \Delta_\theta)$, with volume of $\frac{2}{3}|\Delta_\theta|$. In contrast, when using the horseshoe prior (23), we revert to the usual CLT-based confidence intervals, and the volume of the confidence region converges to zero almost surely.

### S3.2. Proposition 4.2 - Robustness of FAB-PPI under the Horseshoe Prior

Let $\pi_0$ be the horseshoe prior (23), and consider the FAB confidence region $\mathcal{R}_\delta^{\text{FABPP}}(\widehat{\Delta}_\theta; \widehat{\sigma}_\theta)$ for $\Delta_\theta$, as defined in Equation (S33).

We first state a corollary of Cortinovis & Caron (2024, Theorem 3.4), which follows from the power-law tails of the marginal likelihood under the horseshoe prior (see Section S1.1). The corollary states that, if $|\widehat{\Delta}_\theta|$ is very large, then the standard CLT-based confidence interval, $[\widehat{\Delta}_\theta \pm \widehat{\sigma}_\theta z_{1-\delta/2}]$, is recovered.

**Corollary S3.5.** *([Cortinovis & Caron](#) (2024, Theorem 3.4)) For any $\sigma > 0$,*

$$\lim_{\Delta \to \pm\infty} \mathcal{R}_\delta^{FABPP}(\Delta; \sigma) - \Delta = [-\sigma z_{1-\delta/2}, \sigma z_{1-\delta/2}],$$

*where the convergence is with respect to the Hausdorff distance on closed subsets of $\mathbb{R}$.*

Define

$$\mathcal{S}_{\alpha,\delta}(\widehat{\Delta}_\theta, \widehat{\sigma}_\theta, \widehat{m}_\theta, \widehat{\sigma}_\theta^f) = \mathcal{R}_\delta^{\text{FABPP}}(\widehat{\Delta}_\theta; \widehat{\sigma}_\theta) + \mathcal{T}_{\alpha-\delta}(\widehat{m}_\theta; \widehat{\sigma}_\theta^f), \tag{S40}$$

where

$$\mathcal{T}_{\alpha-\delta}(\widehat{m}_\theta; \widehat{\sigma}_\theta^f) = \left[ \widehat{m}_\theta - \widehat{\sigma}_\theta^f z_{1-(\alpha-\delta)/2}, \widehat{m}_\theta + \widehat{\sigma}_\theta^f z_{1-(\alpha-\delta)/2} \right]$$

is the standard CLT-based confidence interval for $m_\theta$. From Corollary S3.5, for any fixed $\sigma > 0$, $m \in \mathbb{R}$, $\sigma^f > 0$,

$$\lim_{\Delta \to \pm\infty} \mathcal{S}_{\alpha,\delta}(\Delta, \sigma, m, \sigma^f) - (\Delta + m) = [-\sigma z_{1-\delta/2} - \sigma^f z_{1-(\alpha-\delta)/2}, \sigma z_{1-\delta/2} + \sigma^f z_{1-(\alpha-\delta)/2}],$$

where, again, the convergence is with respect to the Hausdorff distance on closed subsets of $\mathbb{R}$. Therefore, if $|\widehat{\Delta}_\theta| \gg 0$, the confidence region $\mathcal{S}_{\alpha,\delta}(\widehat{\Delta}_\theta, \widehat{\sigma}_\theta, \widehat{m}_\theta, \widehat{\sigma}_\theta^f)$ reverts to the standard interval

$$[\widehat{\Delta}_\theta + \widehat{m}_\theta \pm (\widehat{\sigma}_\theta z_{1-\delta/2} + \widehat{\sigma}_\theta^f z_{1-(\alpha-\delta)/2})].$$

It follows that, if $\inf_{\theta' \in \mathbb{R}} |\widehat{\Delta}_{\theta'}| \gg 0$, the confidence region for $\theta^\star$, defined as

$$\mathcal{C}_\alpha^{\text{FABPP}} = \left\{ \theta \mid 0 \in \mathcal{R}_\delta^{\text{FABPP}}(\widehat{\Delta}_\theta; \widehat{\sigma}_\theta) + \mathcal{T}_{\alpha-\delta}(\widehat{m}_\theta; \widehat{\sigma}_\theta^f) \right\},$$

reverts to the standard, CLT-based PPI confidence region

$$\mathcal{C}_\alpha^{\text{PP}} = \left\{ \theta \in \mathbb{R} \mid -\widehat{\sigma}_\theta z_{1-\delta/2} - \widehat{\sigma}_\theta^f z_{1-(\alpha-\delta)/2} \leq \widehat{\Delta}_\theta + \widehat{m}_\theta \leq \widehat{\sigma}_\theta z_{1-\delta/2} + \widehat{\sigma}_\theta^f z_{1-(\alpha-\delta)/2} \right\}.$$

### S3.3. Proposition 4.3 - Consistency of FAB-PPI Mean Estimators

Below, we use `BPP` and `BPP+` to distinguish between the estimators $\widehat{\theta}^{\text{FABPP}}, \widehat{\theta}, \widehat{\Delta}$ and $\widehat{\sigma}$ in the two cases of FAB-PPI and FAB-PPI++. Then, the FAB-PPI and FAB-PPI++ mean estimators are given by

$$\widehat{\theta}^{\text{BPP}} = \widehat{\theta}^{\text{PP}} - (\widehat{\sigma}^{\text{PP}})^2 \ell' \left( \widehat{\Delta}^{\text{PP}}; \widehat{\sigma}^{\text{PP}}, \widehat{\sigma}^{\text{PP}} \right),$$

$$\widehat{\theta}^{\text{BPP+}} = \widehat{\theta}^{\text{PP+}} - (\widehat{\sigma}^{\text{PP+}})^2 \ell' \left( \widehat{\Delta}^{\text{PP+}}; \widehat{\sigma}^{\text{PP+}}, \widehat{\sigma}^{\text{PP}} \right),$$

where we recall that $\ell(y; \sigma, \tau) = \log \int_{\mathbb{R}} \mathcal{N}(y; \Delta, \sigma^2) \pi_0(\Delta; \tau) d\Delta$, where $\tau$ is a scale parameter of the prior $\pi_0$. By assumption, both PPI estimators, $\widehat{\theta}^{\text{PP}}$ and $\widehat{\theta}^{\text{PP+}}$, are strongly consistent estimators of $\theta^\star$. It remains to prove that

$$(\widehat{\sigma}^{\text{PP}})^2 \ell' \left( \widehat{\Delta}^{\text{PP}}; \widehat{\sigma}^{\text{PP}}, \widehat{\sigma}^{\text{PP}} \right) \to 0 \tag{S41}$$

$$(\widehat{\sigma}^{\text{PP+}})^2 \ell' \left( \widehat{\Delta}^{\text{PP+}}; \widehat{\sigma}^{\text{PP+}}, \widehat{\sigma}^{\text{PP}} \right) \to 0 \tag{S42}$$

almost surely, as $\min(n, N) \to \infty$. For any $\sigma > 0$, we have $\ell'(y; \sigma, \sigma) = \frac{1}{\sigma} \ell'(y/\sigma; 1, 1)$.

Under the horseshoe prior (23), $\ell'_{\text{HS}}(y; 1, 1)$ is bounded. Therefore, (S41) and (S42) hold almost surely by sandwiching.

Under the Gaussian prior (22),

$$\ell'_{\text{N}}(y; \sigma, \sigma) = -\frac{y}{2\sigma^2}.$$

Hence, since $\widehat{\Delta}^{\text{PP}} \to \Delta$ and $\widehat{\Delta}^{\text{PP+}} \to \Delta$ almost surely, where we recall that $\Delta = \mathbb{E}[f(X) - Y]$, we obtain

$$(\widehat{\sigma}^{\text{PP}})^2 \ell'_{\text{N}} \left( \widehat{\Delta}^{\text{PP}}; \widehat{\sigma}^{\text{PP}}, \widehat{\sigma}^{\text{PP}} \right) \to -\frac{\Delta}{2}$$

$$(\widehat{\sigma}^{\text{PP+}})^2 \ell'_{\text{N}} \left( \widehat{\Delta}^{\text{PP+}}; \widehat{\sigma}^{\text{PP+}}, \widehat{\sigma}^{\text{PP}} \right) \to -\frac{\Delta}{2}$$

almost surely as $\min(n, N) \to \infty$, which implies that the FAB-PPI mean estimators under the Gaussian prior (22) are not consistent.

# S4. Multivariate FAB-PPI

Here we extend FAB-PPI to the multivariate case, where $\theta, m_\theta, \Delta_\theta \in \mathbb{R}^d$. While most of the methodology remains the same as in the univariate case, we now need to specify a multivariate prior for $\Delta_\theta$, for which we consider independent horseshoe priors on each dimension.

## S4.1. Multivariate Bayesian PPI Estimators

As in the univariate case, we use the sample mean $\widehat{m}_\theta$ as the estimator of $m_\theta$. Similarly, we consider some consistent estimator $\widehat{\Delta}_\theta$ of $\Delta_\theta$, such as the sample mean (10), as in PPI, or the control variate estimator (12), as in PPI++. Crucially, we assume that a multivariate CLT holds for this estimator, that is

$$\widehat{\Sigma}_\theta^{-1/2} \left( \widehat{\Delta}_\theta - \Delta_\theta \right) \to \mathcal{N}(0, \mathrm{I})$$

as $\min(n, N) \to \infty$, where $\widehat{\Sigma}_\theta$ is an estimator of $\mathrm{cov}(\widehat{\Delta}_\theta)$. Again, this holds for both the PPI and PPI++ estimators (Angelopoulos et al., 2023a;b). We consider $d$ independent priors, $\pi_0(\Delta_{\theta,k}; \widehat{\sigma}_{\theta,k})$ for $k = 1, \ldots, d$, on the components of $\Delta_\theta$, where $\widehat{\sigma}_{\theta,k}^2$ is the $k$-th diagonal element of $\widehat{\Sigma}_\theta$. The multivariate FAB-PPI estimator $\widehat{\Delta}_\theta^{\mathrm{FABPP}}$ is formed by stacking the individual estimators

$$\widehat{\Delta}_{\theta,k}^{\mathrm{FABPP}} = \widehat{\Delta}_{\theta,k} + \widehat{\sigma}_{\theta,k}^2 \ell' \left( \widehat{\Delta}_{\theta,k}; \widehat{\sigma}_{\theta,k}, \widehat{\sigma}_{\theta,k} \right)$$

for each dimension $k = 1, \ldots, d$. Importantly, note that the $k$-th dimension of $\widehat{\Delta}_\theta^{\mathrm{FABPP}}$ only depends on the $k$-th dimension of the observed $(\mathcal{L}'_\theta(X_i, Y_i) - \mathcal{L}'_\theta(X_i, f(X_i)))$ that are used to estimate $\Delta_{\theta,k}$. The FAB-PPI estimator of $\theta^\star$ then becomes the solution, in $\theta$, to the equation

$$\widehat{m}_\theta + \widehat{\Delta}_\theta^{\mathrm{FABPP}} = \mathbf{0} \in \mathbb{R}^d.$$

## S4.2. Multivariate FAB-PPI Confidence Regions

As in the univariate case, let $\mathcal{T}_{\alpha-\delta}(\widehat{m}_\theta)$ denote a standard $1 - (\alpha - \delta)$ confidence interval for $m_\theta$. For $\Delta_\theta$, we apply the FAB framework with independent horseshoe priors to each dimension $\Delta_{\theta,k}$ and use a union bound to obtain a $1 - \delta$ confidence region for $\Delta_\theta$. In particular, let $\mathcal{R}_{\delta/d}^{\mathrm{FABPP}}(\widehat{\Delta}_{\theta,k}, \widehat{\sigma}_{\theta,k}) = \mathrm{FAB\text{-}CR}(\widehat{\Delta}_{\theta,k}; \pi_0(\cdot\,; \widehat{\sigma}_{\theta,k}), \widehat{\sigma}_{\theta,k}, \delta/d)$ be a $1 - \delta/d$ FAB confidence region for $\Delta_{\theta,k}$ under the horseshoe prior $\pi_0(\cdot\,; \widehat{\sigma}_{\theta,k})$. Then,

$$\mathcal{R}_\delta^{\mathrm{FABPP}}(\widehat{\Delta}_\theta, \widehat{\sigma}_\theta) = \left\{ \Delta_\theta \mid \Delta_{\theta,k} \in \mathcal{R}_{\delta/d}^{\mathrm{FABPP}}(\widehat{\Delta}_{\theta,k}, \widehat{\sigma}_{\theta,k}), \ k = 1, \ldots, d \right\}$$

where $\widehat{\Delta}_\theta = (\widehat{\Delta}_{\theta,1}, \ldots, \widehat{\Delta}_{\theta,d})$, $\widehat{\sigma}_\theta = (\widehat{\sigma}_{\theta,1}, \ldots, \widehat{\sigma}_{\theta,d})$, is a $1 - \delta$ multivariate FAB confidence region for $\Delta_\theta$ by a union bound. With this, the multivariate FAB-PPI confidence region $\mathcal{C}_\alpha^{\mathrm{FABPP}}$ is given by

$$\mathcal{C}_\alpha^{\mathrm{FABPP}} = \left\{ \theta \mid \mathbf{0} \in \mathcal{R}_\delta^{\mathrm{FABPP}}(\widehat{\Delta}_\theta, \widehat{\sigma}_\theta) + \mathcal{T}_{\alpha-\delta}(\widehat{m}_\theta, \widehat{\sigma}_\theta^f) \right\},$$

exactly as in the univariate case. Moreover, also multivariate FAB-PPI enjoys asymptotic coverage as $\min(n, N) \to \infty$. In particular, Theorem 4.1 can be easily extended to the multivariate case by applying a union bound over the dimensions of $\Delta_\theta$.

# S5. Experimental Details

## S5.1. Datasets

Here we provide a brief description of each dataset used for the real data experiments in Section 5.2. For additional details, the reader may refer to Angelopoulos et al. (2023a). All of the datasets were downloaded from the examples provided as part of the `ppi-py` package (Angelopoulos et al., 2023b).

**AlphaFold.** The ALPHAFOLD dataset contains the following features for $N = 10\,802$ protein residues analysed by Bludau et al. (2022): whether the residue is phosphorylated ($Z_i \in \{0,1\}$), whether the residue is part of an intrinsically disordered region (IDR, $Y_i \in \{0,1\}$), and the prediction of the AlphaFold model (Jumper et al., 2021) for the probability of $Y_i$ being

equal to one ($f(X_i) \in [0, 1]$). The goal is to estimate the odds ratio of a protein being phosphorylated and being part of an IDR, i.e.

$$\theta^\star = \frac{\mu_1/(1 - \mu_1)}{\mu_0/(1 - \mu_0)},$$

where $\mu_1 = \Pr(Y = 1 \mid Z = 1)$ and $\mu_0 = \Pr(Y = 1 \mid Z = 0)$. Following Angelopoulos et al. (2023a), given $\alpha \in (0, 1)$, we construct $1 - \alpha/2$ confidence intervals $\mathcal{C}_0 = [l_0, u_0]$ and $\mathcal{C}_1 = [l_1, u_1]$ for $\mu_0$ and $\mu_1$, respectively. Then, by a union bound, the interval

$$\mathcal{C} = \left\{ \frac{c_1}{1 - c_1} \cdot \frac{1 - c_0}{c_0} : c_0 \in \mathcal{C}_0, c_1 \in \mathcal{C}_1 \right\} = \left[ \frac{l_1}{1 - l_1} \cdot \frac{1 - u_0}{u_0}, \frac{u_1}{1 - u_1} \cdot \frac{1 - l_0}{l_0} \right]$$

has coverage at least $1 - \alpha$. Note that the union bound above may result in a conservative confidence interval, leading to coverage significantly larger than $1 - \alpha$ in practice, as in the left panel of Figure 3.

**Forest.** The FOREST dataset contains the following features for $N = 1\,596$ parcels of land in the Amazon rainforest examined during field visits (Bullock et al., 2020): whether the parcel has been subject to deforestation ($Y_i \in \{0, 1\}$) and the prediction of a gradient-boosted tree model for the probability of $Y_i$ being equal to one ($f(X_i) \in [0, 1]$). The goal is to estimate the fraction of Amazon rainforest lost to deforestation, i.e. $\theta^\star = \mathbb{E}[Y]$.

**Galaxies.** The GALAXIES dataset contains the following features for $N = 16\,743$ images from the Galaxy Zoo 2 initiative (Willett et al., 2013): whether the galaxy has spiral arms ($Y_i \in \{0, 1\}$) and the prediction of a ResNet50 model (He et al., 2016) for the probability of $Y_i$ being equal to one ($f(X_i) \in [0, 1]$). The goal is to estimate the fraction of galaxies with spiral arms, i.e. $\theta^\star = \mathbb{E}[Y]$.

**Genes.** The GENES dataset contains the following features for $N = 61\,150$ gene promoter sequences: the expression level of the gene induced by the promoter and the prediction of a transformer model for the same quantity (Vaishnav et al., 2022). The goal is to estimate the median expression level across genes.

**Census.** The CENSUS dataset contains the following features for $N = 380\,091$ individuals from the 2019 California census: the individual's age, sex, and yearly income, as well as the prediction of a gradient-boosted tree model trained on the previous year's raw data for the individual's income. The goal is to estimate the ordinary least squares (OLS) regression coefficients when regressing income on age and sex.

**Healthcare.** The HEALTHCARE dataset contains the following features for $N = 318\,215$ individuals from the 2019 California census: the individual's yearly income and whether they have health insurance ($Y_i \in \{0, 1\}$), as well as the prediction of a gradient-boosted tree model trained on the previous year's raw data for the probability of $Y_i$ being equal to one ($f(X_i) \in [0, 1]$). The goal is to estimate the logistic regression coefficient when regressing health insurance status on income.

### S5.2. Implementation

Code implementing the FAB-PPI method is written in Python and made available at https://github.com/stefanocortinovis/fab-ppi. Comparisons with standard PPI are performed using the ppi-py package (Angelopoulos et al., 2023b). All of the experiments presented here were run locally on an Intel Core i7-11850H CPU.

## S6. Additional Results

### S6.1. Experiments with Synthetic Data

The complete results for the experiments discussed in Section 5 are presented here. The legend names for the figures are as in Section 5.

#### S6.1.1. BIASED PREDICTIONS SIMULATION STUDY

Figure S6 shows the average MSE, CI volume, and CI coverage as a function of the bias level $\gamma$ for the biased predictions study in Section 5.1. Compared to Figure 1, we include results for the non power-tuned methods, as well as for the ones that

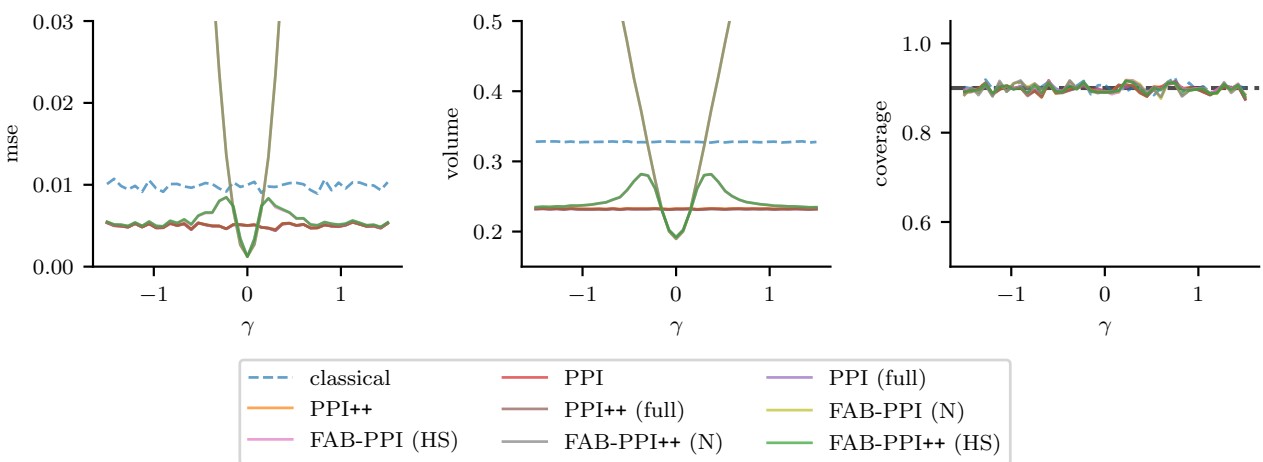

*Figure S6.* Full results for the biased predictions study. The left, middle, and right panels show the average MSE, CI volume, and CI coverage as the bias level $\gamma$ varies.

take into account the uncertainty in the measure of fit $m_\theta$ (i.e. PPI (full) and PPI++ (full)). In this example, power tuning does not play a significant role and the same conclusions as in Section 5.1 hold. In particular, standard PPI induces shorter CIs than classical inference with constant volume across bias levels. On the other hand, FAB methods induce shorter CIs when the predictions are good. As the prediction bias increases, the volume of the FAB CIs with Gaussian prior grows without bound, while the horseshoe prior eventually reverts to the PPI intervals. Furthermore, the coverage plot shows that the methods tested achieve similar coverage to the nominal level and to PPI (full) and PPI++ (full).

### S6.1.2. NOISY PREDICTIONS SIMULATION STUDY

Figure S7 shows the average MSE, CI volume, and CI coverage as a function of $n$ for the values of $\sigma_Y$ considered in the noisy predictions study of Section 5.1. Compared to Figure 2, we include results for the methods that use the Gaussian prior (FAB-PPI (N) and FAB-PPI++ (N)) and those that take into account the uncertainty in the measure of fit $m_\theta$ (i.e. PPI (full) and PPI++ (full)). Like the CI volume plots in the main text, the MSE plots clearly show the benefits of both power tuning and adaptive shrinkage through the horseshoe prior: as $\sigma_Y$ increases, the power-tuned methods clearly outperform the standard alternatives, while shrinkage always helps compared to standard PPI because the predictions remain unbiased. In this case, the Gaussian prior performs similarly to the horseshoe as the prediction rule $f$ is unbiased. The coverage plots confirm that all methods achieve comparable coverage across noise levels.

### S6.2. Experiments with Real Data

#### S6.2.1. MEAN ESTIMATION

**Full Comparison.** Figure S8 shows the average MSE, CI volume, and CI coverage as a function of $n$ for the three datasets considered in Section 5.2. Compared to Figure 3, we include results for the non power-tuned methods, as well as for the ones that take into account the uncertainty in the measure of fit $m_\theta$ (i.e. PPI (full) and PPI++ (full)). The results are consistent with those presented in Section 5.2. In particular, FAB methods outperform the standard PPI alternatives and classical inference, while achieving comparable coverage. For the datasets and the values of $n$ considered, power-tuned methods perform similarly to the non-tuned ones. Among the FAB methods, the horseshoe and Gaussian priors achieve similar performance.

**Example Intervals.** Figure S9 shows 10 randomly chosen intervals for the classical, PPI++, and FAB-PPI++ methods for the three datasets considered in Section 5.2 and different choices of the number of labelled observations $n$.

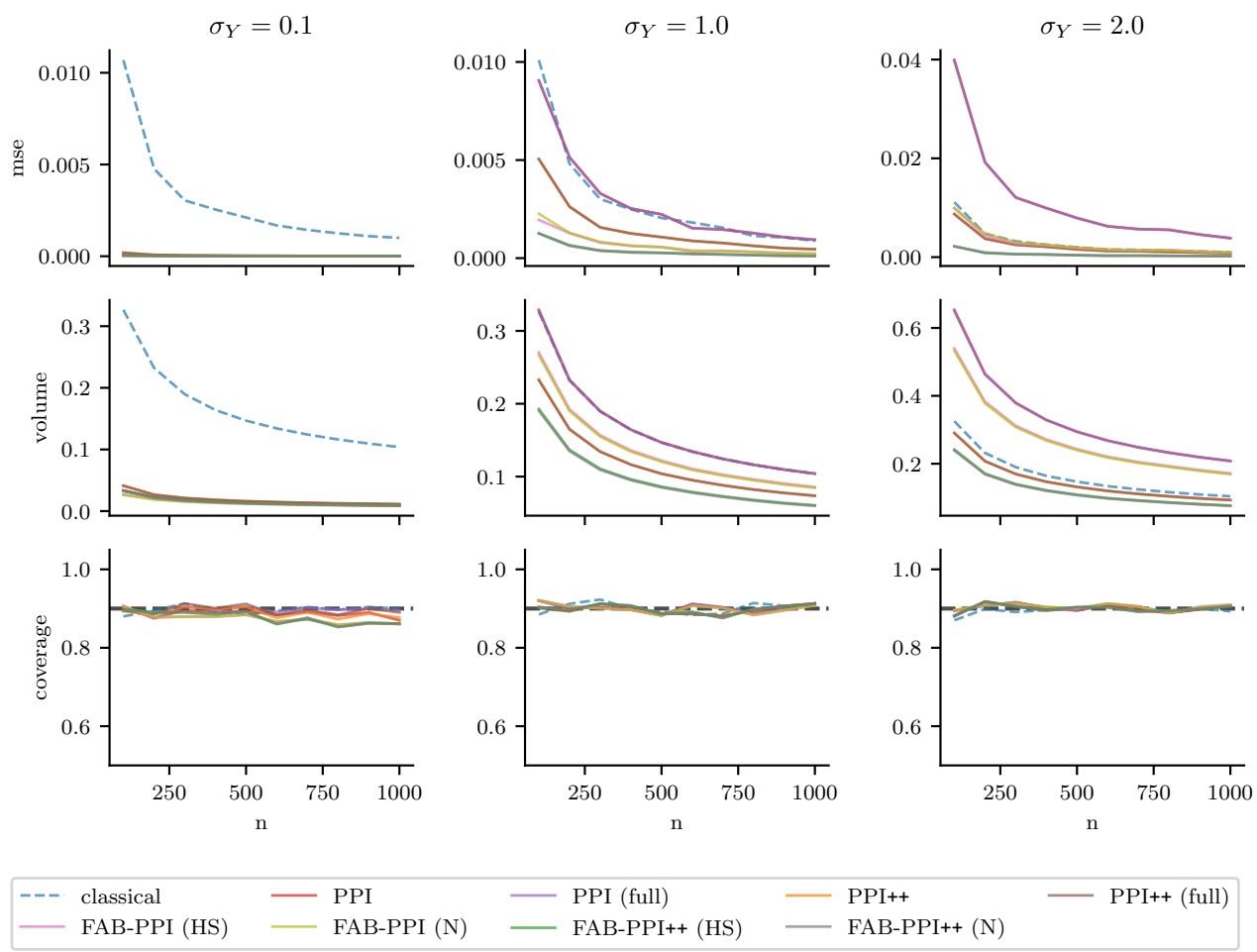

*Figure S7.* Full results for the noisy predictions study. The left, middle, and right panels correspond to noise levels $\sigma_Y = 0.1, 1, 2$, respectively. The top, middle, and bottom rows show average MSE, CI volume, and CI coverage, respectively.

**Varying the Prior Scale.** We repeat the mean estimation experiment on the FOREST dataset while varying the scale of the horseshoe prior used for FAB-PPI++ in Section S6.2.1. In addition to the scale $\widehat{\sigma}$ used in the main text, we consider the sample-independent scale $1/\sqrt{n}$ and the data-independent scale 1. As already mentioned, the computation of the FAB-PPI confidence regions under a horseshoe prior with scale other than $\widehat{\sigma}$ involves numerical integration to compute the corresponding marginal likelihood. Figure S10 shows the average MSE, CI volume, and CI coverage for each of these choices, as well as for classical inference and PPI++. While the scale $\widehat{\sigma}$ achieves the best performance, the other scales also provide shorter CIs than classical inference and PPI++. In particular, the sample independent scale $1/\sqrt{n}$ results in good performance across all metrics without requiring the estimation of $\widehat{\sigma}$.

### S6.2.2. LOGISTIC REGRESSION

Figure S11 shows the average MSE, CI volume, and CI coverage as a function of $n$ for the logistic regression experiment on the HEALTHCARE dataset mentioned in Section 5.2. As mentioned in the main text, FAB methods outperform the standard PPI alternatives and classical inference, while achieving comparable coverage. Among the FAB methods, the horseshoe and Gaussian priors achieve similar performance.

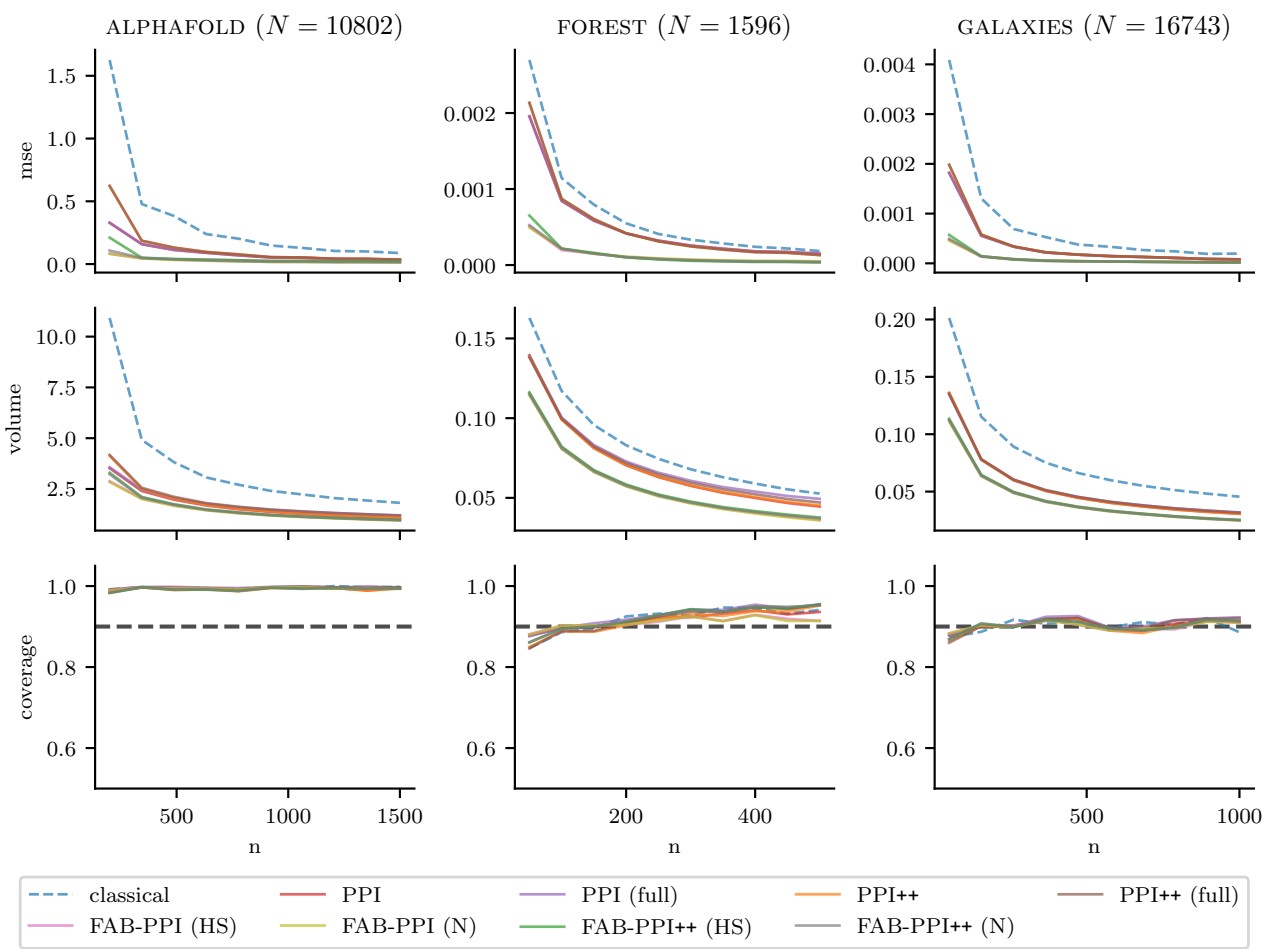

*Figure S8.* Full results for mean estimation experiment on real data. The left, middle, and right panels correspond to the ALPHAFOLD, GALAXIES, and FOREST datasets, respectively. The top, middle, and bottom rows show average MSE, CI volume, and CI coverage, respectively, over 1 000 repetitions for $\alpha = 0.1$.

### S6.2.3. QUANTILE ESTIMATION

Figure S12 shows the average MSE, CI volume, and CI coverage as a function of $n$ for the quantile estimation experiment on the GENES dataset mentioned in Section 5.2. The predictions contained in this dataset are highly biased, and this is reflected in the performance of the FAB-PPI methods. In particular, the Gaussian prior underperforms both classical inference and standard PPI, while the horseshoe prior achieves similar performance to standard PPI thanks to its robustness against large bias levels.

### S6.2.4. LINEAR REGRESSION

Figure S13 shows the average MSE, CI volume, and CI coverage as a function of $n$ for the linear regression experiment on the CENSUS dataset mentioned in Section 5.2. More specifically, panels (a) and (b) correspond to the OLS parameters associated with the *age* and *sex* covariates, respectively. On the one hand, FAB-PPI seems to perform well for the *sex* covariate, with similar performance between the Gaussian and horseshoe priors, and slightly improved MSE and CI volume compared to classical inference and standard PPI. On the other hand, the performance of FAB-PPI for the *age* covariate seems to be affected by bias in the dataset predictions. In particular, FAB-PPI under the Gaussian prior underperforms the alternatives for all $n$. On the other hand, while the horseshoe prior achieves worse performance than the other methods for small $n$, its performance improves as $n$ grows, and it eventually matches standard PPI. This suggests that, as $n$ increases and

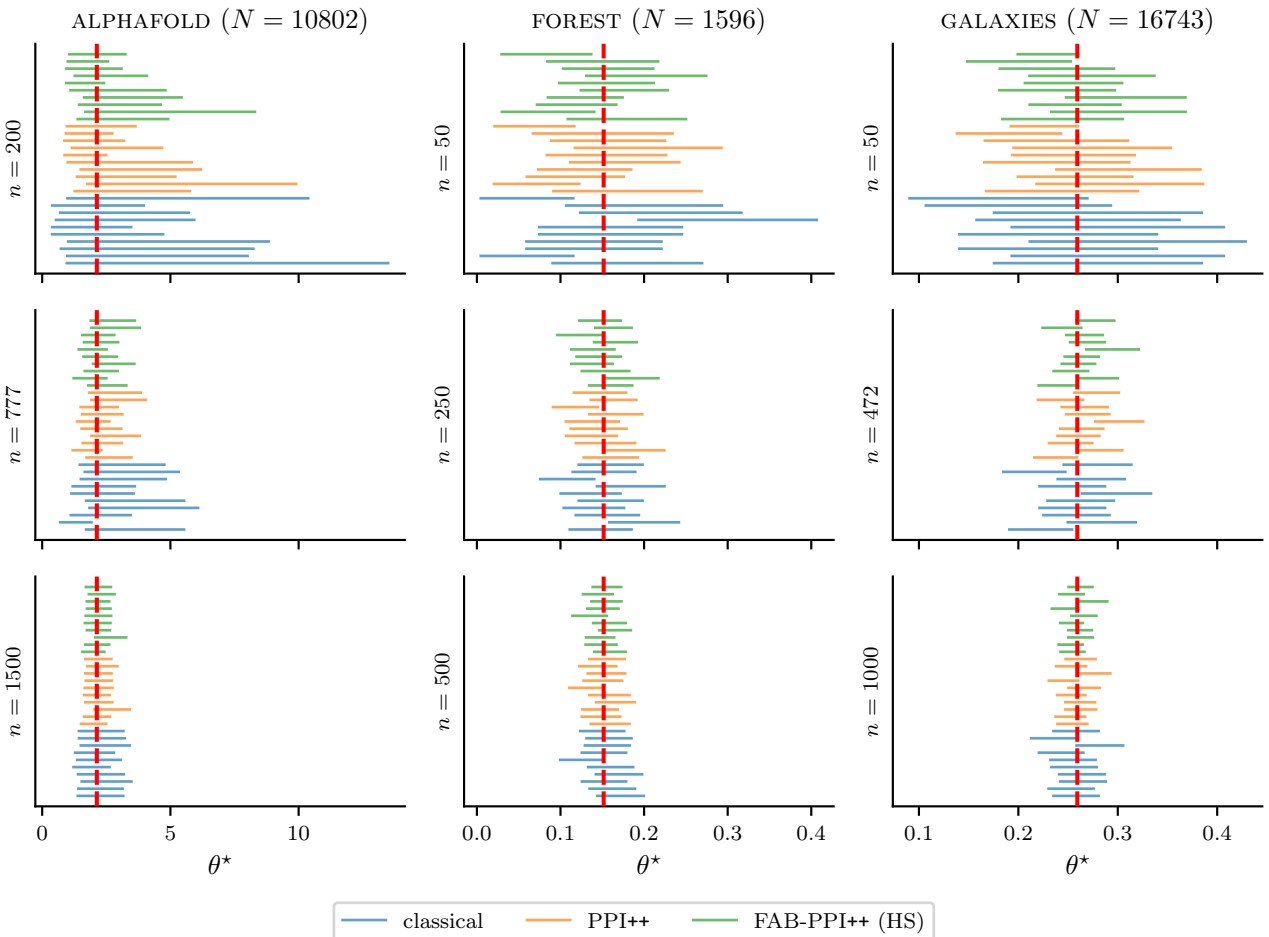

*Figure S9.* Each subfigure includes 10 randomly chosen intervals for the classical, PPI++ and FAB-PPI++ methods. The left, middle, and right panels refer to the ALPHAFOLD, GALAXIES, and FOREST datasets, respectively. The top, middle, and bottom rows correspond to different values of $n$.

$\mathrm{var}(\widehat{\Delta}_\theta)$ decreases, the observed value of the rectifier is increasingly considered as extreme, causing the influence from the horseshoe prior to eventually vanish thanks to its robustness to extreme bias levels.

FOREST $(N = 1596)$

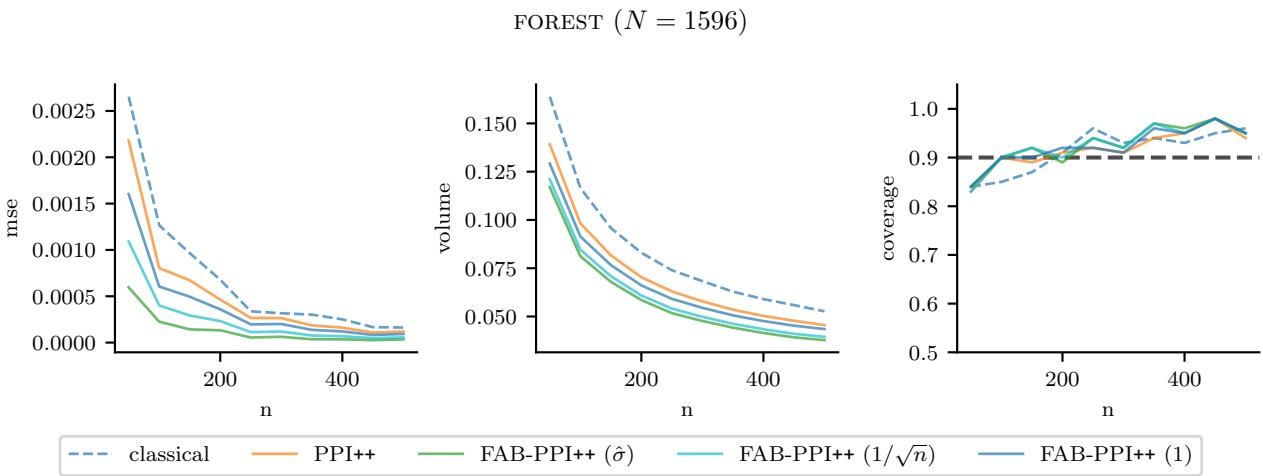

*Figure S10.* Mean estimation experiment on the FOREST dataset with varying horseshoe prior scale. The left, middle, and right panels show average MSE, CI volume, and CI coverage over 100 repetitions for $\alpha = 0.1$.

HEALTHCARE $(N = 318215)$

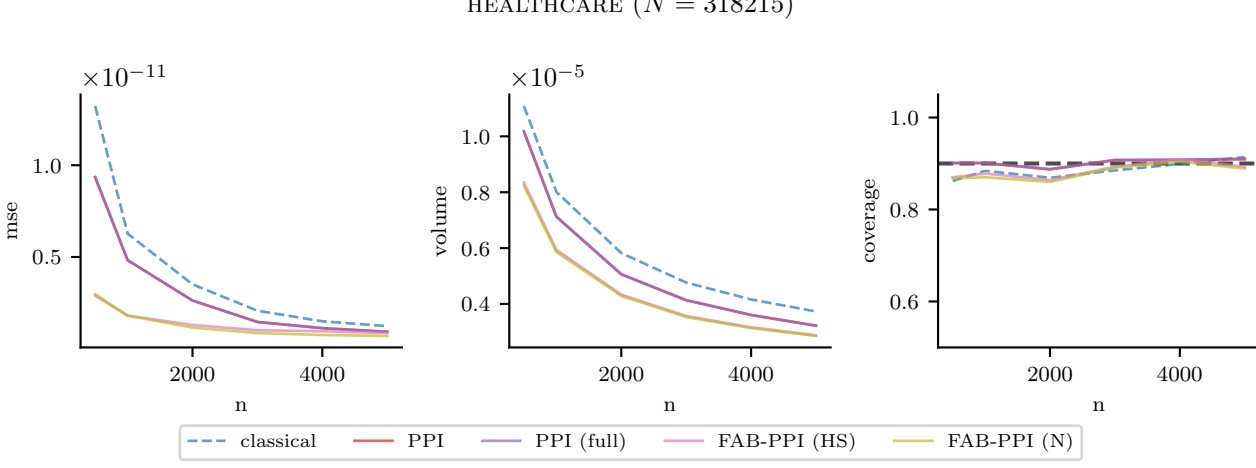

*Figure S11.* Logistic regression experiment on the HEALTHCARE dataset. The left, middle, and right panels show average MSE, CI volume, and CI coverage over 1000 repetitions for $\alpha = 0.1$.

GENES $(N = 61150)$

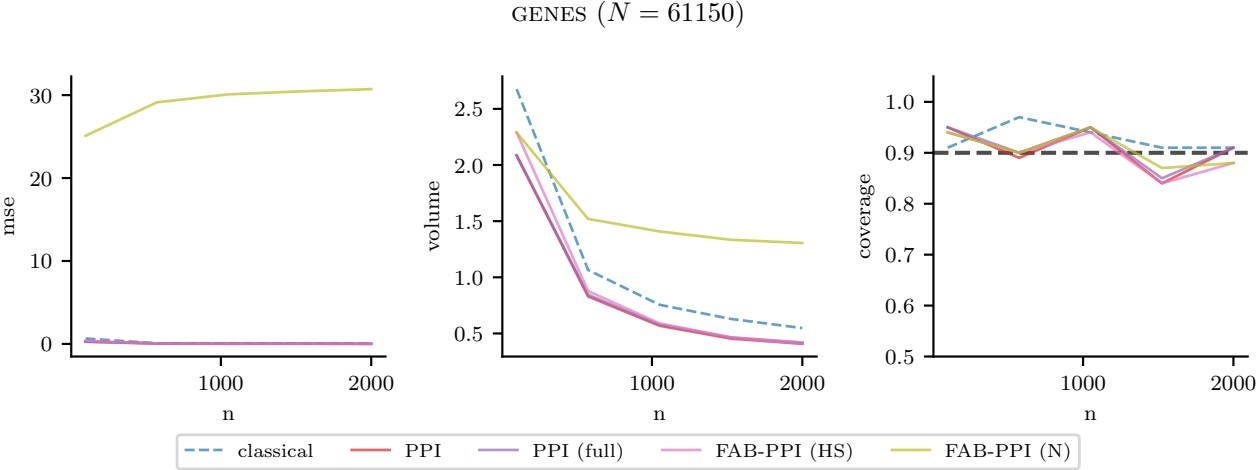

*Figure S12.* Quantile estimation experiment on the GENES dataset. The left, middle, and right panels show average MSE, CI volume, and CI coverage over 100 repetitions for $\alpha = 0.1$.

CENSUS $(N = 380091)$

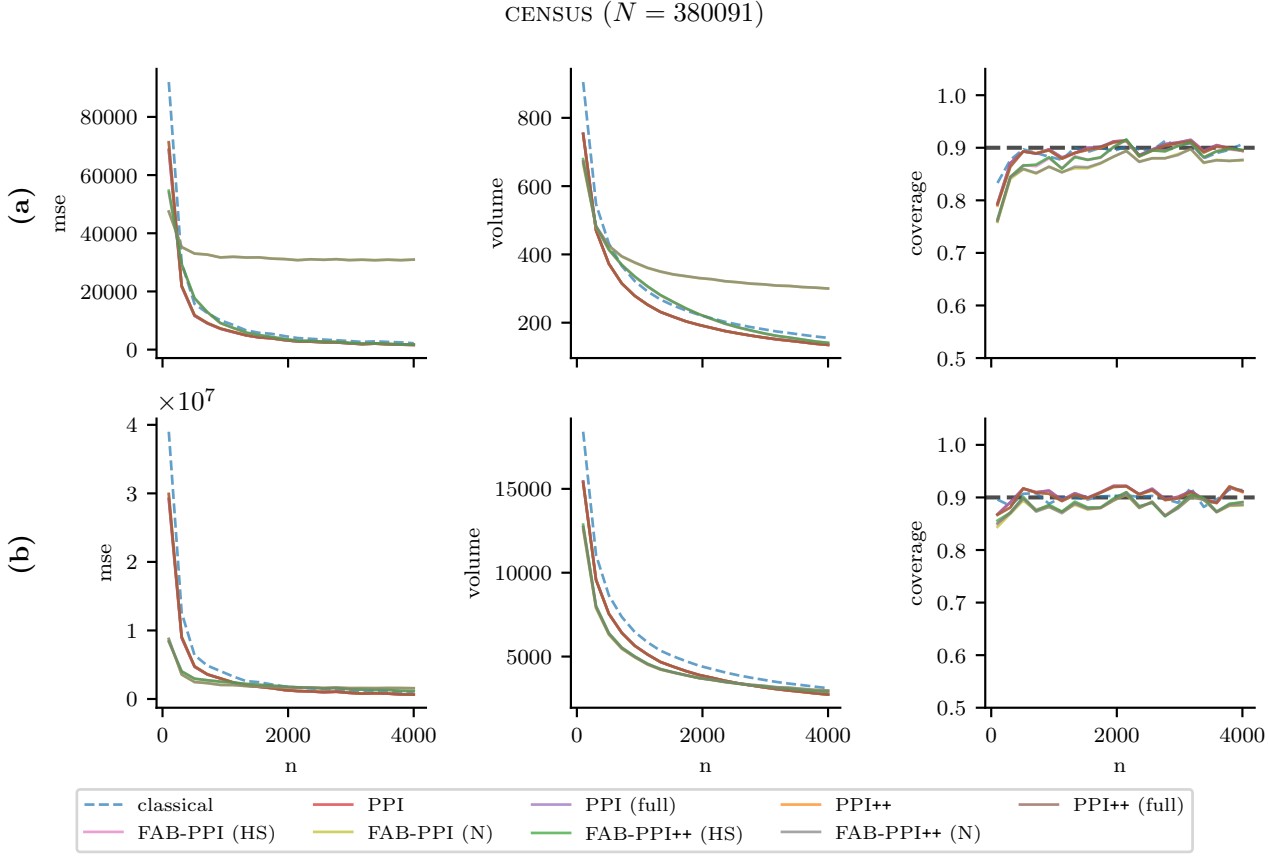

*Figure S13.* Linear regression experiment on the CENSUS dataset. The (a) and (b) panels correspond to the two covariates in the dataset. The left, middle, and right panels show average MSE, CI volume, and CI coverage over 1000 repetitions for $\alpha = 0.1$.

