# OpenReview forum: "FAB-PPI: Frequentist, Assisted by Bayes, Prediction-Powered Inference"
_ICML.cc/2025/Conference — ICML 2025 poster_

### Official Review · Reviewer_KbSV · 2025-03-10

**Overall Recommendation:** 4

**Summary:**

Prediction-powered inference (PPI) improves statistical inference by leveraging machine learning predictions (on unlabeled data) alongside labeled data, resulting in more precise estimates and tighter confidence intervals. The proposed semi-supervised approach, frequentist-assisted-by-Bayes PPI (FAB-PPI), improves upon PPI by incorporating prior knowledge about prediction quality. A horseshoe prior is used to adaptively adjust confidence regions which appears to work well empirically. The approach maintains statistical validity, improves efficiency when predictions are "good", and reverts to standard PPI when predictions are poor/biased, demonstrating in a few real and synthetic data examples its robustness and practical benefits.  The other referenced Bayes-PPI approach gives credible intervals whereas the exposition provides frequentist-type guarantees.

**Claims And Evidence:**

The methods are stated clearly, and the contributions are novel to the best of my knowledge.  The choice of the horseshoe prior is appropriate, and there is discussion (but not experimentation) of other priors that would meet the given desiderata (zero spike, power law tails, analytic).

**Essential References Not Discussed:**

N/A

**Experimental Designs Or Analyses:**

The designs seem appropriate and are both synthetic and real world.  The dimensionality of all datasets is low as the method appears to suffer from CoD (mentioned only at the last sentence of the discussion).

**Methods And Evaluation Criteria:**

The tests are on synthetic data and four low dimensional datasets.  The evaluations appear appropriate (volume levels on synthetic data, mse, volume, and coverage on real data).

**Other Comments Or Suggestions:**

Fig S11. MSE, or deviance?

**Other Strengths And Weaknesses:**

The introduction could be further strengthened by stating up front what characteristics of the predictor f will lead it to be considered "good".

**Questions For Authors:**

1. As above, what comments do you have regarding the coverage level of ~1 for all methods for the first real data set Alphafold (or other coverage levels away from 0.9, some with trend with $n$).

**Relation To Broader Scientific Literature:**

This manuscript provides an approach for frequentist confidence levels in the semi-supervised setting, which is a novel contribution to the best of my knowledge.  Previous relatives included PPI PPI++ and Bayes PPI for credible intervals.  No discussion of how to scale to multivariate analyses limits the approach, but the exposition may provide instruction for future approaches in the SSL framework.

**Theoretical Claims:**

The theoretical claims appear sound.  Technical details are in S2 and S3.

---

> ### Author Rebuttal · Authors · 2025-03-28
>
> We thank the reviewer for evaluating our submission and providing positive and valuable feedback. We are pleased that they recognise our approach’s novelty and clarity.
>
> > The introduction could be further strengthened by stating up front what characteristics of the predictor f will lead it to be considered "good".
>
> In the context of PPI, a "good" predictor $f$ is one that is associated with a value of the rectifier $\Delta_\theta$ close to zero. The rectifier $\Delta_\theta=\mathbb E[\mathcal L_\theta'(X,Y)-\mathcal L_\theta'(X,f(X))]$ is a problem-specific measure of the quality of the predictions of $f$, associated with the specified loss $\mathcal L_\theta$. For example, under the square loss $\mathcal L_\theta(x,y)=\frac{1}{2}(y-\theta)^2$, a "good" predictor $f$ is one such that $f(x)\simeq \mathbb E[Y|X=x]$.
>
> We thank the reviewer for the suggestion, and we will make sure to clarify this in the revised manuscript right after we define $\Delta_\theta$.
>
> > Fig S11. MSE, or deviance?
>
> In `Figure S11`, like elsewhere in the paper, we use the MSE as the measure of estimation error to ensure consistency across all experiments. We will make sure to state this clearly in the revised manuscript.
>
> > As above, what comments do you have regarding the coverage level of ~1 for all methods for the first real data set Alphafold (or other coverage levels away from 0.9, some with trend with $n$).
>
> The task performed on the Alphafold dataset involves the estimation of an odds ratio, defined as
> $$
>     \theta^\ast = \frac{\mu_1/(1 - \mu_1)}{\mu_0/(1 - \mu_0)},
> $$
> where $\mu_0$ and $\mu_1$ are the means of two groups in the dataset. FAB-PPI applies to estimands that may be expressed as minimisers of a convex loss, as defined in `Eq. (1)`. However, this is not the case for $\theta^\ast$ above, preventing the standard application of FAB-PPI to obtain confidence regions for $\theta^\ast$. To overcome this, we construct $1-\alpha/2$ confidence regions for $\mu_0$ and $\mu_1$ separately and merge them into a confidence interval for $\theta^\ast$ through a union bound. The resulting interval is guaranteed to have coverage at least $1 - \alpha$, but in practice it is often too conservative. That is, the high coverage observed in `Figure 3` is not due to a failure of FAB-PPI (or of classical inference and standard PPI, for that matter), but rather is a limitation of the union bound construction of the CI for $\theta^\ast$. Nonetheless, since the same union bound strategy is used for all the methods (including classical inference and PPI) presented in `Figure 3`, comparing the volume of the resulting CIs is still informative.
>
> This procedure to construct valid CIs for $\theta^\ast$ was used in [1, Section 3.1 of the arXiv version] and is summarised in `Appendix S5.1`. However, in `Section 5.2` we state that the experiment on the Alphafold dataset involves mean estimation, which is imprecise and may understandably cause confusion when interpreting the coverage in `Figure 3`. We thank the reviewer for raising this point, which we will clarify in the revised manuscript.
>
> Other fluctuations in empirical coverage away from the nominal level include uncercoverage for small $n$ in some experiments and overcoverage as $n$ increases for the Forest dataset. The first phenomenon is due to the fact that all the confidence regions compared (classical, PPI, and FAB-PPI versions) are based on the CLT, as detailed in `Assumption 3.1`. In practice, even when the number of unlabelled samples $N$ is sufficiently large as in our experiments, all methods require the number of labelled samples $n$ to be large enough for the CLT on $\widehat\Delta_\theta$ to kick in and ensure the right coverage level, while small $n$ may lead to fluctuations in the coverage level for all the methods considered. To mitigate this, one could alternatively employ (non-uniform) Berry-Esseen bounds (see e.g. [2] and references within) to obtain more conservative asymptotically valid CIs for both PPI and FAB-PPI.
>
> On the other hand, the seemingly increasing trend in coverage for the Forest dataset when $n > 200$ is due to the size of the dataset, which is the smallest considered ($N = 1596$). In particular, the "ground truth" against which the empirical coverage is computed is the sample mean of $Y$ across the labels of the entire dataset. As the size $n$ of the labelled subset used for PPI grows, there is a non-negligible overlap between the latter and the full dataset used to compute the ground truth (e.g. overlap $>30\\%$ when $n = 500$), explaining the overcoverage. Once again, this issue applies to all methods considered, and the comparison of the volume of the resulting confidence regions remains informative.
>
> [1] Angelopoulos, A. N., Bates, S., Fannjiang, C., Jordan, M. I., and Zrnic, T. (2023). Prediction-Powered Inference. arXiv preprint arXiv:2301.09633.
>
> [2] M. Austern and L. Mackey (2022).  Efficient concentration with Gaussian approximation. arXiv:2208.09922.

---

### Official Review · Reviewer_QSTy · 2025-03-12

**Overall Recommendation:** 4

**Summary:**

The authors propose a new scheme for combining experimental data with model predictions, effectively for inferring valid confidence intervals for estimators where some of the samples are noisier due to being model predictions. The work extends recent prediction-power inference methods by encoding prior knowledge on model errors, in practice a horseshoe prior, and provides detailed theoretical derivation and analysis of the estimator.

**Claims And Evidence:**

There are clearly itemised claims that are backed by both theoretical and empirical evidence. The claimed improvements (validity, robustness and efficiency) are relevant and have value for the community, and especially for people applying PPI in scientific research in other fields.

**Essential References Not Discussed:**

None

**Experimental Designs Or Analyses:**

There is clear synthetic data experiment that helps understanding how the approach works, also illustrating how the CIs would behave under a worse prior choice (Gaussian). The other experiments are reasonable, but somewhat harder to interpret. FAB-PPI is shown to have smaller CI, but there is no direct empirical evaluation of whether the CI is correct. The Supplement S6 seems to provide the missing quantification in terms of coverage -- maybe this could be already in the main paper?

One weakness is that the artificial experiment setups are extremely simplified (which is good for understanding the idea, but they do not really help in understanding how well it works for higher-dimensional problems etc), but the real-world examples in Sec 5.2 complement them well. The results are again primarily from the perspective of CI volume reduction, but now coverage is explicitly evaluated as well. For Alphafold the coverage does not appear to be correct but instead remains roughly around 1 for all $n$, which contradicts the main claims to some extent. There is some discussion on this and the comparison method PPI++ also fails so the issue seems to be more related to the overall family of PPI methods, but the discussion is not sufficiently deep.

**Methods And Evaluation Criteria:**

The proposed method is sound and it is evaluated according to valid criteria. Adding prior information on model errors makes sense and the horseshoe prior encodes a reasonable inductive bias. The choice of the specific prior is discussed in Section 4.3 and the choice made in this work is justified sufficiently well, e.g. by explaining why some simpler alternatives would not be ideal.

**Other Comments Or Suggestions:**

None.

**Other Strengths And Weaknesses:**

The work has high potential for impact in other fields. PPI is recent high-profile technique that many application areas are likely already considering, and theoretically justified improvements for PPI are valuable for them. There is a possibility this becomes a de-facto solution for the specific task at hand.

**Questions For Authors:**

1. Could you open up more the somewhat negative result for Alphafold, where you have clearly too high coverage for all PPI variants. Why does it happen, what are the implications, and could we somehow recognise this failure more in practical use?

**Relation To Broader Scientific Literature:**

The paper builds very extensively on Angelopoulos et al. (2023a,2023b), for instance only citing these two sources (in addition to one quick remark on applications) during the first page of the paper. Other related work is properly described later, but nevertheless the work feels somewhat incremental improvement over what Angelopoulos et al. already provided -- the scientific idea of adding a prior is fairly obvious and the theoretical results to an extent follow from previous PPI works. This is not a major limitation as the PPI idea is already getting traction (100+ citations since 2023) and the approach has clear uses across broad range of sciences, but still a minor weakness.

**Theoretical Claims:**

I have somewhat limited expertise on the kind of theoretical analysis done here and did not manage to evaluate the proofs in detail, but overall the analysis aligns with the claims and I did not observe any major gaps or errors. The theoretical analysis could be considered as the main contribution of the work, given the somewhat obvious initial idea, and hence the overall merits of the paper somewhat depend on how significant these contributions are in isolation. That said, even if the development was completely straightforward the paper would have value.

---

> ### Author Rebuttal · Authors · 2025-03-28
>
> We would like to thank the reviewer for their time in evaluating our submission and for providing positive and valuable feedback. We are glad that the reviewer appreciates the potential impact of our work.
>
> > The Supplement S6 seems to provide the missing quantification in terms of coverage -- maybe this could be already in the main paper?
>
> We thank the reviewer for the suggestion. We will make sure to include coverage plots for the synthetic data experiments in the main text of the revised manuscript.
>
> > Could you open up more the somewhat negative result for Alphafold, where you have clearly too high coverage for all PPI variants. Why does it happen, what are the implications, and could we somehow recognise this failure more in practical use?
>
> The task performed on the Alphafold dataset involves the estimation of an odds ratio, defined as
> $$
>     \theta^\ast = \frac{\mu_1/(1 - \mu_1)}{\mu_0/(1 - \mu_0)},
> $$
> where $\mu_0$ and $\mu_1$ are the means of two groups in the dataset. FAB-PPI applies to estimands that may be expressed as minimisers of a convex loss, as defined in `Eq. (1)`. However, this is not the case for $\theta^\ast$ above, preventing the standard application of FAB-PPI to obtain confidence regions for $\theta^\ast$. To overcome this, we construct $1-\alpha/2$ confidence regions for $\mu_0$ and $\mu_1$ separately and merge them into a confidence interval for $\theta^\ast$ through a union bound. The resulting interval is guaranteed to have coverage at least $1 - \alpha$, but in practice it is often too conservative. That is, the high coverage observed in `Figure 3` is not due to a failure of FAB-PPI (or of classical inference and standard PPI, for that matter), but rather is a limitation of the union bound construction of the CI for $\theta^\ast$. Nonetheless, since the same union bound strategy is used for all the methods (including classical inference and PPI) presented in `Figure 3`, comparing the volume of the resulting CIs is still informative.
>
> This procedure to construct valid CIs for $\theta^\ast$ was used in [1, Section 3.1 of the arXiv version] and is summarised in `Appendix S5.1` of our paper. However, in `Section 5.2` we state that the experiment on the Alphafold dataset involves mean estimation, which is imprecise and may understandably cause confusion when interpreting the coverage in `Figure 3`. We thank the reviewer for raising this point, which we will clarify in the revised manuscript.
>
> [1] Angelopoulos, A. N., Bates, S., Fannjiang, C., Jordan, M. I., and Zrnic, T. (2023). Prediction-Powered Inference. arXiv preprint arXiv:2301.09633.

---

### Official Review · Reviewer_nWdf · 2025-03-17

**Overall Recommendation:** 4

**Summary:**

This paper proposes a Bayesian adaptation of Prediction-powered inference (PPI) problem. PPI is a method which provides confidence interval and estimators in the presence of predictions of (black-box) machine learning models and small amount of labels. This work allows for the possibility of a prior distribution over predictions indicating their quality. The authors demonstrate that this method improves over a pure frequentist PPI, and defaults to a standard PPI when the prior over predictions is a horshoe prior.

**Claims And Evidence:**

The claims made in the introduction and methodolgy seems to be well supported by theory. The paper is well written and easy to follow.

**Essential References Not Discussed:**

From my knowledge of this area, essential references seems to be discussed well.

**Experimental Designs Or Analyses:**

The experiments are conducted on both synthetic and real data. The baselines considered makes sense (a PPI without prior, classical estimator) and the authors demonstrate the superior performance of the approach, especially when the number of samples is low, where use of prior is more justified.

**Methods And Evaluation Criteria:**

The proposed method seems to be plausible and definitely interesting for the problem. The consistency of the resulting estimators with different priors like Horseshoe and Gaussian prior are discussed. The evaluation criteria is sound with multiple real world datasets. Overall I find the presented approach and experiments very pertinent to the problem.

**Other Comments Or Suggestions:**

NA

**Other Strengths And Weaknesses:**

The paper is well written and easy to follow. The idea of using prior over predictions is novel and interesting. The authors also show the consistency of the estimators under various priors, which is valuable.

**Questions For Authors:**

Although usage of prior is interesting, the question I have is if the prior is not accurate (i.e. the estimate of the quality of the predictions is not accurate), then how would the performance of the proposed FAB-PPI get affected?

**Relation To Broader Scientific Literature:**

I am not familiar too much with PPI, so it is hard to assert the significance of the proposed method with respect to wider literature. However, from the context of PPI as a method, prior augmented PPI looks like a welcome contribution that is useful in its own right.

**Theoretical Claims:**

I have not checked for proofs in detail.

---

> ### Author Rebuttal · Authors · 2025-03-28
>
> We would like to thank the reviewer for their time in evaluating our submission and for providing positive and valuable feedback. We are glad that the reviewer found the paper well written and easy to follow.
>
> > Although usage of prior is interesting, the question I have is if the prior is not accurate (i.e. the estimate of the quality of the predictions is not accurate), then how would the performance of the proposed FAB-PPI get affected?
>
> As stated in the paper, FAB-PPI is motivated by applications in which the black-box predictor $f$ is expected to be of high quality for the given estimation problem, as measured by the rectifier $\Delta_\theta$. When this is the case, FAB-PPI results in smaller confidence regions and more accurate estimates than standard PPI. To achieve this, FAB-PPI specifies a prior distribution centered at zero for $\Delta_\theta$. In this setting, the chosen prior is not accurate if the true $\Delta_\theta$ is not actually close to zero. In the `Biased prediction` experiment in `Section 5.1`, we investigate the behaviour of FAB-PPI as we vary the accuracy of the prior, which is controlled by $\gamma$ (the higher $|\gamma|$, the farther away $\Delta_\theta$ from zero, and, hence, the less accurate the prior). In this case, the performance of FAB-PPI depends on the specific prior chosen, as well as on the accuracy level $\gamma$. Under the Gaussian prior, as $|\gamma|$ increases, the resulting FAB-PPI confidence regions grow unbounded, eventually becoming larger than both the classical and PPI confidence intervals. Conversely, under the horseshoe prior, as $|\gamma|$ increases, the resulting confidence regions also become larger initially, but they remain bounded, eventually reverting to the standard PPI confidence intervals. In this sense, this property of FAB-PPI under the horseshoe prior, which actually holds for any prior with power-law tails, may be seen as a form of robustness to prior mispecification. The behaviour of FAB-PPI under the Gaussian and horseshoe priors as $\gamma$ varies is illustrated in `Figure 1` of the paper.
>
> Additionally, note that, while in the paper we use the scale $\sigma$ of the noise of the generative model as the prior scale $\tau_n$ to obtain a parameter-free approach, in practice one could partially control the behaviour of FAB-PPI in the presence of prior inaccuracy through the choice of the prior scale $\tau_n$. In particular, let $|\bar\gamma|$ be the accuracy level in `Figure 1` at which the FAB-PPI confidence region under the chosen prior becomes larger than the standard PPI confidence interval. Then, a larger $\tau_n$ would imply that the prior gives relatively more weight to moderately high values of $\Delta_\theta$, thereby increasing $|\bar\gamma|$ (i.e. leading to an improvement of FAB-PPI over PPI across a wider range of accuracy levels). However, the cost of increasing $\tau_n$ is that the potential gains when $\Delta_\theta \approx 0$ become smaller (i.e. the size of the improvement of FAB-PPI over PPI for $\gamma \approx 0$ decreases), as the prior gives relatively less weight to very small values of $\Delta_\theta$.

---

### Decision · Program_Chairs · 2025-05-01

**Decision:**

Accept (poster)

**Comment:**

The authors propose an adaptation of the recent PPI technique which incorporates a Bayesian-like prior on the accuracy of the input prediction function f, resulting in confidence regions that remain valid but are tighter for settings in which f is accurate (which correspond to most application settings of interest).   The paper is very clearly written; the authors analyze the consequences of choosing either a Gaussian or horseshoe prior, and nicely illustrate their methods with both synthetic experiments and several real data sets studied in the prior work.  Authors were unanimous in recommending acceptance.